# Optimizing AVR system performance via a novel cascaded RPIDD$^2$-FOPI controller and QWGBO approach

Serdar Ekinci[1], Václav Snášel[2], Rizk M. Rizk-Allah[2,3], Davut Izci[1,4,5]*, Mohammad Salman[6]*, Ahmed A. F. Youssef[6]

1 Department of Computer Engineering, Batman University, Batman, Turkey, 2 Faculty of Electrical Engineering and Computer Science, VŠB-Technical University of Ostrava, Ostrava, Czechia, 3 Basic Engineering Science Department, Menoufia University, Al Minufiyah, Egypt, 4 Applied Science Research Center, Applied Science Private University, Amman, Jordan, 5 MEU Research Unit, Middle East University, Amman, Jordan, 6 College of Engineering and Technology, American University of the Middle East, Egaila, Kuwait

* mohammad.salman@aum.edu.kw (MS); davutizci@gmail.com, davut.izci@batman.edu.tr (DI)

**Data Availability Statement:** The authors confirm that the data of this study is available within the manuscript.

## Abstract

Maintaining stable voltage levels is essential for power systems' efficiency and reliability. Voltage fluctuations during load changes can lead to equipment damage and costly disruptions. Automatic voltage regulators (AVRs) are traditionally used to address this issue, regulating generator terminal voltage. Despite progress in control methodologies, challenges persist, including robustness and response time limitations. Therefore, this study introduces a novel approach to AVR control, aiming to enhance robustness and efficiency. A custom optimizer, the quadratic wavelet-enhanced gradient-based optimization (QWGBO) algorithm, is developed. QWGBO refines the gradient-based optimization (GBO) by introducing exploration and exploitation improvements. The algorithm integrates quadratic interpolation mutation and wavelet mutation strategy to enhance search efficiency. Extensive tests using benchmark functions demonstrate the QWGBO's effectiveness in optimization. Comparative assessments against existing optimization algorithms and recent techniques confirm QWGBO's superior performance. In AVR control, QWGBO is coupled with a cascaded real proportional-integral-derivative with second order derivative (RPIDD$^2$) and fractional-order proportional-integral (FOPI) controller, aiming for precision, stability, and quick response. The algorithm's performance is verified through rigorous simulations, emphasizing its effectiveness in optimizing complex engineering problems. Comparative analyses highlight QWGBO's superiority over existing algorithms, positioning it as a promising solution for optimizing power system control and contributing to the advancement of robust and efficient power systems.

## Introduction

### Background

Maintaining nominal voltage levels within electrical power systems is a fundamental requirement for ensuring the efficient and reliable performance of the entire power infrastructure [1–3]. The deviation in terminal voltage of alternators during load changes represents a significant

**Funding:** The author(s) received no specific funding for this work.

**Competing interests:** The authors have declared that no competing interests exist.

challenge, with consequences ranging from equipment damage to operational disruptions and costly downtime [4]. To address this critical issue, automatic voltage regulators (AVRs) [5] are employed, offering control over generator exciter output power to regulate the terminal voltage magnitude [6].

## Literature review

In the landscape of AVR control, controllers play a pivotal role in monitoring and regulating the AVR itself [7]. These controllers enable real-time adjustments for voltage stability, facilitate remote monitoring, fault detection, and automatic shutdown during emergencies, thereby enhancing system dependability [8]. Diverse controllers are available, from standard proportional-integral-derivative (PID) to advanced variants like PID acceleration (PIDA), fractional-order PID (FOPID), and PID with a second-order derivative (PIDD²) [9–16]. The choice of a cost function is equally significant, affecting performance significantly. Researchers employ various objective functions, such as the integral of time-weighted squared error, integral of squared error, integral of absolute error, and the dynamic response performance criteria-based Zwe-Lee Gaing [17,18].

Various research studies have explored the application of metaheuristic optimization algorithms to determine optimal controller gains in AVR systems [19]. For instance, in [20], the study enhances AVR performance using the adaptive neuro-fuzzy inference system (ANFIS) and compares it with scenarios without a controller and with a PID controller. The ANFIS, trained with a hybrid optimization learning scheme, shows improved performance metrics in MATLAB/Simulink simulations. In [21], the self-competitive differential evolution (DE) algorithm is introduced, enhancing exploration ability with a control parameter. Applied to real-world optimization problems, including PID controller tuning for AVR systems, the self-competitive DE outperforms other DE algorithms. The study in [22] focuses on optimizing a controller for a real AVR system in a synchronous generator, proposing a novel proportional-integral controller with anti-windup protection. The African vultures optimization algorithm (AVOA) is adaptively modified for controller parameter design, resulting in superior AVR controller performance. In [23], the equilibrium optimizer (EO) is employed to optimize parameters for a novel AVR controller using a time-domain objective function. Comparative analyses with various controllers validate the proposed controller's superior performance in settling time, rise time, and overshoot, supported by frequency domain analysis. The enhanced whale optimization algorithm (EWOA) in [24] stabilizes PID controller parameters in AVR systems. Comparative analysis establishes EWOA's faster convergence, higher precision, shorter execution time, and greater stability, making it a practical method for PID controller optimization. The study in [25] introduces a novel $PI^{\lambda 1}I^{\lambda 2}D^{\mu 1}D^{\mu 2}$ controller for AVR system, optimized using the mayfly algorithm. Comparative analyses demonstrate the proposed controller's excellence in both time and frequency domain analyses, as well as robustness and disturbance rejection. In [26], the marine predator optimization algorithm (MPA) optimizes a FOPID controller for AVR systems. Comparative analyses highlight MPA–FOPID's superior stability, frequency response, robustness, response speed, and disturbance-rejection capabilities. The study in [7] models the AVR system as a sextuple-input single-output system, optimizing controller parameters using the particle swarm optimization African vultures optimization algorithm (PSO–AVOA). Comparative analyses showcase the effectiveness of the proposed design technique and optimization algorithm. In [27], FOPID controller optimization using the salp swarm algorithm (SSA), ant lion optimization (ALO), and PSO is assessed for AVR system. Comparative analysis demonstrates promising early results and emphasizes the superiority of the proposed controller. The study in [28] also introduces the MPA for

tuning the FOPID controller in AVR system, showing exceptional performance in enhancing AVR transient response compared to other FOPID controllers optimized with recent meta-heuristic algorithms. In [29], an AVR system with a PID controller addresses issues related to dynamic changes in power systems. The zebra optimization algorithm (ZOA) and osprey optimization algorithm (OOA) optimize transient response, with ZOA showing significant improvement. The study in [30] combines a $PIDND^2N^2$ controller with the balanced arithmetic optimization algorithm (b-AOA) to enhance AVR system stability. The proposed approach excels in transient response and frequency response, outperforming state-of-the-art control methods. In [31], the distance and Levy-flight based crow search algorithm (DLCSA) optimizes FOPID, FOPI, and FOPD controllers for AVR systems, showcasing superior performance in various aspects compared to established techniques.

## Research gap and motivation

The use of metaheuristic optimization algorithms to tune controllers has improved the performance of many AVR systems. However, there is still a concern about their precision, prompting the need for further improvement. Therefore, introducing new controllers and integrating state-of-the-art metaheuristic optimization algorithms has the potential to enhance accuracy and overall performance in AVR systems. In response to these constraints, our study aims to pioneer novel approaches to AVR control, contributing to the advancement of robust and efficient power systems. The primary objective is to introduce an advanced control scheme capable of addressing these limitations effectively. To achieve this, we have developed a novel optimizer, designed to fine-tune and optimize the parameters of our proposed control scheme, enhancing overall performance and adaptability.

## Challenges

While current control methodologies [32–39] have shown promise, their limitations necessitate a more nuanced approach. Challenges include issues related to robustness, overshoots, rise times, settling times, and persistent steady-state errors.

## Contribution

Our study introduces the quadratic wavelet-enhanced gradient-based optimization (QWGBO) algorithm, which serves as an innovative tuning mechanism to enhance the gradient-based optimization (GBO) [40]. GBO, grounded in the principles of gradient concepts from Newton's rules, demonstrates robust local exploration capabilities and parameter simplicity. However, it faces challenges when dealing with complex, high-dimensional optimization problems. The QWGBO algorithm integrates two crucial enhancements, the quadratic interpolation mutation (QIM) and the wavelet mutation strategy (WMS), which are combined with GBO to improve exploration and exploitation capabilities. QIM enhances exploration by approximating the objective function at candidate positions, diversifying the search process. Meanwhile, WMS addresses stagnation issues and improves solution accuracy, exploring new solution vectors using wavelet functions with adjustable parameters. The fusion of these components forms the QWGBO algorithm, offering an effective mechanism to enhance GBO's optimization capabilities, striking a balance between exploration and exploitation.

The efficacy of the QWGBO is first verified through statistical and non-parametric Wilcoxon signed rank tests using widely adopted unimodal, multimodal, and fixed-dimensional multimodal benchmark functions. We present the experimental results obtained by comparing the proposed QWGBO algorithm with other competitive and recent optimization algorithms, original gradient-based optimization [40], gravitational search algorithm [41], whale

optimization algorithm [42], slime mould algorithm [43], prairie dog optimization [44]. The results clearly demonstrate the superior performance of the QWGBO algorithm in solving the benchmark functions, as it consistently achieves lower minimum values and ranks first in all comparisons indicating its potential as an effective optimization algorithm.

Our work presents an innovative approach that combines both the controller and the optimizer to create a comprehensive solution for enhancing AVR stability. The core innovation is the QWGBO algorithm, the efficacy of which is demonstrated through extensive statistical and non-parametric tests on benchmark functions, showing its superior optimization capabilities. Our study introduces a cascaded real $PIDD^2$ ($RPIDD^2$) and fractional-order proportional-integral (FOPI) controller, designed for precision and stability in voltage regulation, and fine-tuned by QWGBO for improved performance and adaptability. The work targets the minimization of dynamic response performance criteria using Zwe-Lee Gaing objective function [45], ensuring that the AVR system meets stringent performance requirements.

To validate the proposed approach's superiority, extensive comparative analyses (statistical, boxplot, convergence profile, Wilcoxon signed-rank test, transient and frequency responses, performance against varying input reference and external load disturbance, controller effort and robustness) were conducted against other competitive algorithms and recently reported optimization techniques. In addition to the comparative assessment against the algorithms used in the above analyses, further comparisons were made with recently reported 17 optimization techniques in the literature [23,25,31,46–59]. The simulation results unequivocally highlight the QWGBO algorithm's superior performance in optimizing the AVR system, as evident from lower objective function values, excellent convergence, and statistical assessments as well as stability and robustness analyses. This work thus contributes to the advancement of control systems for power infrastructure, with the QWGBO algorithm emerging as a promising solution to optimize complex engineering problems. In light of the above presentation, the contributions of this study can be listed as follows.

- The study introduces the QWGBO algorithm, innovatively enhancing the GBO. QWGBO integrates quadratic interpolation mutation and wavelet mutation strategy to improve exploration and exploitation capabilities, addressing challenges faced by GBO in complex, high-dimensional optimization problems.

- The effectiveness of QWGBO is rigorously verified through statistical and non-parametric tests using widely adopted unimodal, multimodal, and fixed-dimensional multimodal benchmark functions. Comparative analyses against competitive optimization algorithms demonstrate QWGBO's superior performance.

- The work presents an innovative approach by combining the QWGBO with a cascaded $RPIDD^2$-FOPI controller. This comprehensive solution enhances AVR stability, targeting the minimization of dynamic response performance criteria and ensuring compliance with stringent performance requirements.

- The study introduces a cascaded $RPIDD^2$-FOPI controller, designed for precision and stability in voltage regulation. QWGBO is employed for fine-tuning, resulting in improved performance and adaptability. The proposed approach minimizes dynamic response performance criteria using the Zwe-Lee Gaing objective function, ensuring effective AVR system performance.

- The proposed approach is rigorously validated through extensive comparative analyses, including statistical tests, boxplot comparisons, convergence profiles, Wilcoxon signed-rank tests, transient and frequency responses, performance against varying input references and

external load disturbances, controller effort analysis, and robustness assessments. The analyses encompass a wide range of competitive algorithms and recently reported optimization techniques in the literature.

- The simulation results unequivocally demonstrate the superior performance of the QWGBO algorithm in optimizing the AVR system. Comparative assessments against a diverse set of optimization techniques showcase lower objective function values, excellent convergence, and superior stability and robustness analyses. This contribution advances control systems for power infrastructure, positioning QWGBO as a promising solution for optimizing complex engineering problems.

## Paper organization

The paper is organized into several sections to systematically present the proposed QWGBO approach and its application in enhancing the stability of AVR system. The structure is as follows. The next section provides the basics of the original GBO. The third section presents the proposed QWGBO approach. The performance evaluation of the proposed QWGBO against benchmark functions is presented in the fourth section. The structure of the AVR system is explained in the fifth section. The sixth section describes the new methodology for the proposed in this work in detail, including the new cascaded RPIDD$^2$-FOPI controller, objective function and constraints of optimization problem. The application of QWGBO in optimizing the proposed controller parameters is also outlined in this section. The seventh section presents and discusses the simulation results obtained from the application of QWGBO to the AVR system. Finally in the eighth section, the paper concludes with a summary of findings and potential directions for future research.

## Basics of gradient-based optimization

The gradient-based optimization (GBO) represents one of the effective metaheuristic algorithms that was introduced by Ahmadianfar [40] based on the gradient concepts of the Newton's rules. In terms of optimization viewpoint, the GBO was established based on two searching operators which are the gradient searching rule (GSR) which promotes to improve the exploration search possibility, and local escaping operator (LEO) which emphasizes the exploitation search pattern. This approach is successfully investigated on 28 benchmark functions involving different optimization characteristics and six engineering structural designs, where the results of GBO have provided promising performance over the other counterparts. Since Ahmadianfar introduced GBO in 2020, it has drawn the attention of scientists and engineers across a variety of fields on a global scale including the structural optimization problems [60], optimal power flow problems [61], photovoltaic models [62], feature selection [63], directional overcurrent relay problem [64]. The main procedures of the GBO is described in the subsequent sections and its core framework is displayed using the pseudocode provided in Algorithm 1.

```
Algorithm 1. The core framework of the GBO.
1: Insert the algorithm parameters: N,T,pr,ε,t = 0
2: Initialize a population of random positions, Z_i^t = {Z_{i,1}^t, Z_{i,1}^t, ..., Z_{i,D}^t}, i ∈ N
3: Compute the fitness function of each solution, f(Z_i^t), i ∈ N
4 Obtain the best and worst solutions, Z_best^t and Z_worst^t
5 While t≤T do
4:    for n = 1:N
5:      for j = 1:D
```

```
 6:        Create random integer indices from the population size,
           r1≠r2≠r3≠r4≠n
 7:        Update the position of Zₙᵗ⁺¹ using (13)
 8:     end for
 9:     if rand<pr
10:       Calculate the position of the LEO phase (Zᵗ_LEO) using (15)
11:            Zₙᵗ⁺¹ = Zᵗ_LEO
12:       end if
13:       Update Zᵗ_best and Zᵗ_worst
14:     end for
15: t = t+1
19: end while
20: Output: return Zᵗ_best
```

## Initialization

Like several optimization methods, GBO starts with a population of solutions that are randomly generated within the search space limits as follows.

$$Z_n = Z_{min} + rand \times (Z_{max} - Z_{min}), n = 1, 2, \ldots, N \tag{1}$$

where *rand* stands for a random vector between [0,1] generated according to uniform distribution, N refers to the population size, and $Z_{max}$ and $Z_{min}$ indicate the upper and lower limits of the search domain.

## The GSR operator (Exploration)

The GSR is the key element of the GBO since it can encourage exploration capability while avoiding the local optimal tramping problem. GSR uses a numerical gradient approach rather than a direct function derivation to advance the initial guess to the next place. Additionally, the direction of movement (DM) is included to take advantage of the immediate vicinity of the current solution. Thus, the following equation can be adopted to update the position of the current vector ($Z_n^t$).

$$Z1_n^t = Z_n^t - randn \times B_1 \times \frac{2\Delta Z.Z_n^t}{(Z_{worst} - Z_{best} + \varepsilon)} + rand \times B_2 \times \left(Z_{best} - Z_n^t\right) \tag{2}$$

where $Z1_n^t$ defines the renewed vector, the second part of Eq (2) realize the GSR process while the third part defines the DM which aims to guide the current solution towards the best so for solution ($Z_{best}$); $\varepsilon$ defines a small number inside the interval [0, 0.1], $t$ stands for the current iteration, $Z_{worst}$ defines the worst solution obtained so far, *rand* and *randn* define random values that are generated according to uniform and normal distributions, respectively. Here, transition parameter ($B_1$) is defined with the intention of balancing the search process' exploration and exploitation and it can be expressed as follows.

$$B_1 = 2.rand.a - a \tag{3}$$

$$a = \left| G.sin\left(\frac{3\pi}{2} + sin\left(G.\frac{3\pi}{2}\right)\right) \right| \tag{4}$$

$$G = G_{min} + (G_{max} - G_{min}).\left(1 - \left(\frac{t}{T}\right)^3\right)^2 \tag{5}$$

where and $T$ stands for the maximum size of iterations, and $G_{max}$ and $G_{min}$ are set to 1.2 and 0.2, respectively. Morevere, the parameter $B_2$ is another component in GBO aims to support the exploration process which is defined as follows.

$$B_2 = 2.rand.a - a \tag{6}$$

$$\Delta Z = rand(1:D).|step| \tag{7}$$

$$step = \frac{(Z_{best} - Z_{r1}^t) + \gamma}{2} \tag{8}$$

$$\gamma = 2.rand.\left(\left|\frac{Z_{r1}^t + Z_{r2}^t + Z_{r3}^t + Z_{r4}^t}{4} - Z_n^t\right|\right) \tag{9}$$

where $step$ sigifies the step length that is conied using the positions of the $Z_{best}$ and $Z_{r1}^t$; $rand(1:D)$ defines a vector of random values with $D$ dimensions, and $r1{:}r4$ define different integers arbitrarily conined from the population size, where $(r1{\neq}r2{\neq}r3{\neq}r4{\neq}n)$. By regarding the best solution so far ($Z_{best}$) instead of the present solution vector ($Z_n^t$) in (2), an updated solution vector ($Z2_n^t$) can be created as follows:

$$Z2_n^t = Z_{best} - randn \times B_1 \times \frac{2\Delta Z.Z_n^t}{(yp_n^m - yq_n^m + \varepsilon)} + rand \times B_2 \times \left(Z_{r1}^t - Z_{r2}^t\right) \tag{10}$$

$$yp_n = rand.\left(\frac{[Y_n^t + Z_n^t]}{2} + rand.\Delta Z\right) \tag{11}$$

$$yq_n = rand.\left(\frac{[Y_n^t + Z_n^t]}{2} - rand.\Delta Z\right) \tag{12}$$

where $Y_n^t$ defines the define the previous solutions, where if "Flag" is equal to 1, the solution $Y_n^t$ is assigned the value of $Z1_n^t$, otherwise, it is assigned to $Z2_n^t$. The searching stage based on Eq (10) can assist the local searching phase but it isnot usfull for global searching process. In this sense, Eq (2) can realize the global search, but it limits the local search. In order to balance the tendencies of exploration and exploitation patterns, it is beneficial to use both $Z1_n^t$ and $Z2_n^t$ search tactics. In light of this, the new location at iteration ($Z_n^{t+1}$) can be stated as follows.

$$Z_n^{t+1} = r_a \times (r_b \times Z1_n^t + (1 - r_b) \times Z2_n^t) + (1 - r_a) \times Z3_n^t \tag{13}$$

$$Z3_n^m = Z_n^t - \rho_1 \times (Z2_n^t - Z1_n^t) \tag{14}$$

## The LEO operator (Exploitation)

The LEO stage intends to enhance the searching efficacy of the GBO during the iterative process. The LEO can produce an improved performance ($Z_{LEO}^t$) by combining certain solutions which include the solutions $Z1_n^t$ and $Z2_n^t$, two arbitrary alternatives ($Z_{r1}^t$ and $Z_{r2}^t$), the best so far position ($Z_{best}$), and a fresh, randomly selected solution ($Z_k^t$). The solution $Z_{LEO}^t$ can be coined using the following scheme.

 **if** $rand{<}p$**if** $rand{<}0.5$

$$Z^t_{LEO} = Z^{t+1}_n + f_1 \times (v_1 \times Z_{best} - v_2 \times Z^t_k) + f_2 \times B_1 \times (v_3 \times (Z2^t_n - Z1^t_n) + v_2 \times (Z^t_{r1} - Z^t_{r2}))/2$$

$$Z^{t+1}_n = Z^t_{LEO}$$

***else***

$$Z^t_{LEO} = Z_{best} + f_1 \times (v_1 \times Z_{best} - v_2 \times Z^t_k) + f_2 \times \rho_1 \times (v_3 \times (Z2^t_n - Z1^t_n) + v_2 \times (Z^t_{r1} - Z^t_{r2}))/2$$

$$Z^{t+1}_n = Z^t_{LEO} \tag{15}$$

end
end

where $f_1$ stands for arbitrary number drawn according to the uniform distribution from [–1,1], while $f_2$ denotes another arbitrary number drawn according to the normal distribution with zero mean and standard deviation of 1, $pr$ is the switching probability. The parameters $v_1$, $v_2$, and $v_3$ are expressed as follows.

$$v_1 = \begin{cases} 2 \times rand & if \ \vartheta_1 < 0.5 \\ 1 & otherwise \end{cases} \tag{16}$$

$$v_2 = \begin{cases} rand & if \ \vartheta_1 < 0.5 \\ 1 & otherwise \end{cases} \tag{17}$$

$$v_3 = \begin{cases} rand & if \ \vartheta_1 < 0.5 \\ 1 & otherwise \end{cases} \tag{18}$$

where $rand$ stands for random number drawn according to the uniform distribution and ranged from 0 to 1, and $\vartheta_1$ defines a number inside the interval [0,1]. The forementioned equations of the $v_1$, $v_2$, and $v_3$ can be simplified with following aspects.

$$v_1 = H_1 \times 2 \times rand + (1 - H_1) \tag{19}$$

$$v_2 = H_1 \times rand + (1 - H_1) \tag{20}$$

$$v_3 = H_1 \times rand + (1 - H_1) \tag{21}$$

where $H_1$ takes a binary value (0 or 1). In this regard, if $\vartheta_1 < 0.5$, then $H_1$ is 1, otherwise, it is 0. The solution $Z^t_k$ in (15) is updated based on the following prospective.

$$Z^t_k = \begin{cases} Z_{rand} & if \ \vartheta_2 < 0.5 \\ Z^t_p & otherwise \end{cases} \tag{22}$$

$$Z_{rand} = Z_{min} + rand(0, 1) \times (Z_{max} - Z_{min}) \tag{23}$$

where $Z_{rand}$ defines randomly generated solution, $Z^t_p$ stands for randomly selected solution from the current population ($p \in N$), and $\vartheta_2$ indicates a random number inside the interval

[0,1]. Eq (22) can be reformulated as follows.

$$Z_k^m = H_2 \times Z_p^t + (1 - H_2) \times Z_{rand} \tag{24}$$

where $H_2$ takes a binary value (0 or 1). If $\vartheta_2 < 0.5$, then $H_2$ is 1, otherwise, it is 0.

## The integrated QWGBO approach

Although the GBO has its unique characteristics and strengths, it also has some shortages. The GBO provides some outstanding local exploration capabilities with easy control parameters, however it may be prone to stuck in local optima that is not supportive for the global optimal solution while tackling with complicated optimization and high-dimensional issues. Therefore, combining the GBO with improvement strategies is a prudent way to enhance the exploration and exploitation searches while mitigating the stuck in local optima dilemma. In this regard, the original GBO is integrated with quadratic interpolation mutation (QIM) and Wavelet mutation strategy (WMS) to improve the search efficiency of the original GBO, which is named QWGBO algorithm. In the proposed framework of QWGBO, the GBO starts the iterative searching phase using its own optimization operators to explore the entire search space. On the other hand, the QIM is appended to further enhance the exploration phase and then the WMS is incorporated to direct the search towards better vicinities as well as averting the stagnation dilemma. By this perspective, it is intended to enhance the searchability of the algorithm and boost its accuracy. The QWGBO based on the presented strategies can be demonstrated as follows.

## The concept of QIM phase

The QIM phase is appended in the searching process of the GBO to enhance the exploration pattern while detecting new regions within the search space which leads to effective diversity of the presented algorithm in terms of space quality. The QIM employs an approximate polynomial interpolation method by utilizing known function values at specific candidate positions to create a low order interpolating polynomial that closely approximates the original objective function. Subsequently, the obtained outcome of the polynomial is utilized as a close approximation to the candidate solution of the given function. Hence, this can generate more approximated solutions to the original ones, enhancing diversity during the algorithm's search process. To be specific, two surrounding solutions $(i+1)^{th}$, and $(i+2)^{th}$ for the current one ($i^{th}$ individual) are selected, then the quadratic function ($Q(Z)$) of each one is formulated, which is the approximation for the exact function. Let $Q(Z)$ defines the quadratic interpolating polynomial at abscissae $Z_i$, $Z_{i+1}$, $Z_{i+2}$, and it is defined as follows.

$$\begin{cases} Q(Z_i) = a_0 + a_1 Z_i + a_2 Z_i^2 = f(Z_i) \\ Q(Z_{i+1}) = a_0 + a_1 Z_{i+1} + a_2 Z_{i+1}^2 = f(Z_{i+1}) \\ Q(Z_{i+2}) = a_0 + a_1 Z_{i+2} + a_2 Z_{i+2}^2 = f(Z_{i+2}) \end{cases} \tag{25}$$

The smallest value of the approximate quadratic function $Q(Z)$ can be reached at

$$Z^* = \frac{1}{2} \times \frac{(Z_{i+1}^2 - Z_{i+2}^2)f(Z_i) + (Z_{i+2}^2 - Z_i^2)f(Z_{i+1}) + (Z_i^2 - Z_{i+1}^2)f(Z_{i+2})}{(Z_{i+1} - Z_{i+2})f(Z_i) + (Z_{i+2} - Z_i)f(Z_{i+1}) + (Z_i - Z_{i+1})f(Z_{i+2})} \tag{26}$$

Using this strategy, a more refined solution can be acquired, demonstrating a better fitness value than the previous solution. Additionally, the updating procedure can be carried out in a

greedy fashion as follows.

$$\begin{cases} Z_i^{t+1} = Z_i^t & f(Z_i^t) \leq f(Z^*) \\ Z_i^{t+1} = Z^* & f(Z_i^t) > f(Z^*) \end{cases} \tag{27}$$

Once the QIM is applied to each individual, the population's quality can be enhanced.

## The concept of WMS phase

To mitigate the stagnation phenomena of the original GBO and improve the accuracy of solution during the iterative optimization process, wavelet mutation strategy (WMS) is purported to guide the search towards enriched areas. WMS has the ability to explore to detect new vectors of solutions surrounding the chosen solutions within the feasible space using the expansion and translation features of the wavelet function. Furthermore, the wavelet function's stretching parameters can be adjusted to lower the function's amplitude, which adjusts the range of mutation with the progress of iterations. Thus, WMS is used in place of the original mutation technique. To be specific, for the chosen solution $Z_i = (Z_{i1}, Z_{i2}, \ldots, Z_{iD})$ determined by mutation probability $Pm$, the corresponding updated solution can be expressed as follows.

$$\begin{cases} Z_{i,j}^{t+1} = Z_{i,j}^t + p_1 & r < 0.5 \\ Z_i^{t+1} = Z_{i,j}^t + p_2 & r \geq 0.5 \end{cases} \tag{28}$$

where $p_1$ and $p_2$ stand for the wavelet mutation operators which are expressed by

$$\begin{cases} p_1 = \sigma.(Z_{max,j} - Z_{i,j}^t) \\ p_2 = \sigma.(Z_{i,j}^t - Z_{min,j}) \end{cases} \tag{29}$$

where $r$ defines a random value in [0,1]; denotes the value of wavelet function that can be described as follows.

$$\sigma = \frac{1}{\sqrt{a}}\psi\left(\frac{\varphi}{a}\right) \tag{30}$$

$$\psi(y) = e^{-y^2/2}.cos(5y) \tag{31}$$

where $\varphi$ denotes a random parameter selected randomly from the interval $[-2.5a, 2.5a]$, $\psi$ defines the Morlet wavelet function, and $a$ denotes the stretching parameter that is expressed as follows.

$$a = s.\left(\frac{1}{s}\right)^{\left(1-\frac{t}{T}\right)} \tag{32}$$

where $s$ signifies a random value within the range [800, 1200]. By this way, the algorithm can search in a larger space during the early stages and perform a slight mutation in the later stages of the search, which improves the searchability of the algorithm. The framework of the suggested QWGBO is depicted as in Fig 1.

## Performance evaluation on benchmark functions

### Details of used benchmark functions

In this section, we provide a brief overview of the benchmark functions used in the experimentation. These benchmark functions serve as test cases to evaluate the performance of optimization algorithms. The following benchmark functions are considered.

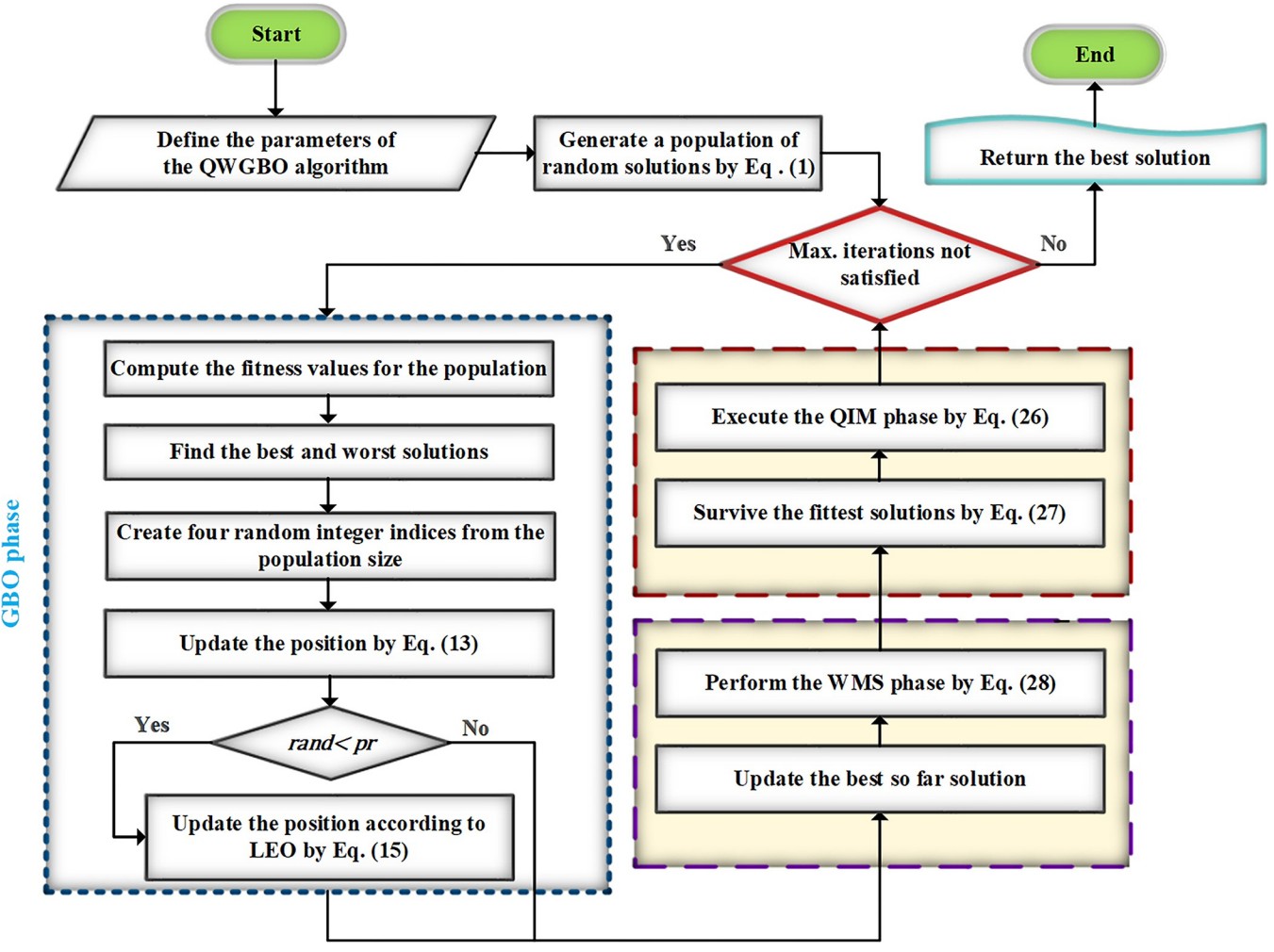

**Fig 1. The working scheme of the QWGBO approach.**

**Step Function ($BF_1$):** Step Function is a unimodal function with the optimal value of 0. It is defined as follows.

$$BF_1 = \sum_{i=1}^{D} (x_i + 0.5)^2 \tag{33}$$

- Function Dimension (Number of Variables), D: 30

- Lower Bound for Each Variable, LB: -100

- Upper Bound for Each Variable, UB: 100

**Penalized2 Function ($BF_2$):** Penalized2 Function is a multi-modal function with the optimal value of 0. The Penalized2 Function is defined as follows.

$$BF_2 = 0.1\left\{ sin^2(3\pi x_1) + \sum_{i=1}^{D-1} (x_i - 1)^2[1 + sin^2(3\pi x_{i+1})] + (x_D - 1)^2[1 + sin^2(2\pi x_D)]\right\} + \sum_{i=1}^{D} u(x_i, 5, 100, 4) \tag{34}$$

- Function Dimension (Number of Variables), D: 30

- Lower Bound for Each Variable, LB: -50

- Upper Bound for Each Variable, UB: 50

**Foxholes Function ($BF_3$):** Foxholes Function is a fixed-dimension multi-modal function with an optimal value of 0.998. The Foxholes Function is defined as follows.

$$BF_3 = \left( \frac{1}{500} + \sum_{j=1}^{25} \frac{1}{j + \sum_{i=1}^{2} (x_i - a_{ij})^6} \right)^{-1} \tag{35}$$

- Function Dimension (Number of Variables), D: 2

- Lower Bound for Each Variable, LB: -65.536

- Upper Bound for Each Variable, UB: 65.536

**Shekel 7 Function ($BF_4$):** Shekel 7 Function is a fixed-dimension multi-modal function with an optimal value of -10.4029. The Shekel 7 Function is defined as follows.

$$BF_4 = -\sum_{i=1}^{7} \frac{1}{(X - a_i)(X - a_i)^T + c_i} \tag{36}$$

- Function Dimension (Number of Variables), D: 4

- Lower Bound for Each Variable, LB: 0

- Upper Bound for Each Variable, UB: 10

These benchmark functions are commonly used in optimization research to assess the performance of optimization algorithms and compare their efficiency in finding optimal solutions. The optimal values provided for each function represent the minimum value that an optimization algorithm should aim to find.

## Experimental results on benchmark functions

In this section, we present the experimental results obtained by comparing the proposed QWGBO algorithm with other competitive and recent optimization algorithms, including original gradient-based optimization (GBO) [40], gravitational search algorithm (GSA) [41], whale optimization algorithm (WOA) [42], slime mould algorithm (SMA) [43], prairie dog optimization (PDO) [44]. The goal of this analysis is to assess the performance of the QWGBO algorithm on a set of benchmark functions, and to demonstrate its superior performance. The experimental setup for the comparisons included the following parameters: 500 total iterations, a population size of 30, and 30 independent runs for each algorithm to obtain statistically meaningful results.

In Table 1, the results of the benchmark functions are presented in terms of the minimum, maximum, average, standard deviation, median, and rank for each algorithm. The functions evaluated include the Step Function, Penalized2 Function, Foxholes Function, and Shekel 7 Function. For the Step Function, QWGBO achieved the lowest minimum value of 0, indicating its superior ability to find the optimal solution. It outperformed all other algorithms, including GBO, GSA, WOA, SMA, and PDO. The rank of 1 demonstrates the highest overall

**Table 1. Comparative statistical results obtained against benchmark functions.**

| Function | Algorithms | Minimum | Maximum | Average | Standard deviation | Median | Rank |
|---|---|---|---|---|---|---|---|
| $BF_1$ | QWGBO | **0** | **2.1276E−19** | **7.0920E−21** | **3.8844E−20** | **0** | **1** |
| | GBO | 5.6685E−07 | 2.1777E−05 | 5.3662E−06 | 6.4390E−06 | 2.1667E−06 | 2 |
| | GSA | 8.3122E−17 | 1.9981E−02 | 8.2673E−04 | 3.7231E−03 | 2.4790E−16 | 3 |
| | WOA | 6.0468E−02 | 9.7266E−01 | 3.8197E−01 | 2.6025E−01 | 3.0013E−01 | 5 |
| | SMA | 1.1248E−03 | 2.0315E−02 | 7.1312E−03 | 5.2822E−03 | 6.0131E−03 | 4 |
| | PDO | 9.4421E−01 | 7.2500E+00 | 3.2228E+00 | 1.6418E+00 | 2.8094E+00 | 6 |
| $BF_2$ | QWGBO | **1.5705E−32** | **8.3786E−25** | **5.5857E−26** | **2.1257E−25** | **1.5705E−32** | **1** |
| | GBO | 1.7365E−08 | 1.2494E−06 | 3.2991E−07 | 2.9145E−07 | 2.6445E−07 | 2 |
| | GSA | 1.0458E−01 | 4.1678E+00 | 1.8016E+00 | 9.7786E−01 | 1.7285E+00 | 6 |
| | WOA | 2.0927E−03 | 6.7231E−02 | 2.1335E−02 | 1.4959E−02 | 1.8088E−02 | 4 |
| | SMA | 3.3893E−06 | 2.1846E−02 | 5.4851E−03 | 6.5595E−03 | 2.2762E−03 | 3 |
| | PDO | 9.7328E−03 | 1.5970E+00 | 3.5923E−01 | 4.5696E−01 | 1.6273E−01 | 5 |
| $BF_3$ | QWGBO | **1.3498E−32** | **1.4653E−15** | **1.3439E−16** | **3.2192E−16** | **1.3498E−32** | **1** |
| | GBO | 4.0104E−06 | 5.4792E−02 | 1.4159E−02 | 1.7461E−02 | 1.0995E−02 | 3 |
| | GSA | 9.4290E−02 | 2.6130E+01 | 8.2813E+00 | 7.0093E+00 | 6.0551E+00 | 6 |
| | WOA | 1.6932E−01 | 1.0371E+00 | 4.4282E−01 | 2.4585E−01 | 4.2944E−01 | 4 |
| | SMA | 8.3576E−04 | 3.2922E−02 | 9.3062E−03 | 8.1392E−03 | 6.5686E−03 | 2 |
| | PDO | 2.0368E+00 | 3 | 2.9319E+00 | 2.3280E−01 | 2.9994E+00 | 5 |
| $BF_4$ | QWGBO | **−1.0403E+01** | **−1.0403E+01** | **−1.0403E+01** | **0** | **−1.0403E+01** | **1** |
| | GBO | **−1.0403E+01** | −2.7659E+00 | −8.3119E+00 | 2.8517E+00 | −1.0403E+01 | 4 |
| | GSA | **−1.0403E+01** | −2.8612E+00 | −9.9290E+00 | 1.8075E+00 | −1.0403E+01 | 3 |
| | WOA | **−1.0403E+01** | −1.8371E+00 | −7.7802E+00 | 3.1226E+00 | −1.0370E+01 | 5 |
| | SMA | **−1.0403E+01** | −1.0401E+01 | −1.0402E+01 | 6.2972E−04 | −1.0403E+01 | 2 |
| | PDO | −1.0400E+01 | −1.9905E+00 | −4.9030E+00 | 2.4317E+00 | −4.4865E+00 | 6 |

performance. In the case of Penalized2 Function, QWGBO once again obtained the lowest minimum value of 1.5705E−32, showcasing its superior performance in minimizing the objective function. It outperformed all other algorithms, securing the top rank of 1. For Foxholes Function, QWGBO excelled by achieving the lowest minimum value of 1.3498E−32, which is significantly better than the other algorithms. It ranked 1, indicating its dominant performance on this function. Lastly, for the Shekel 7 Function, QWGBO obtained a minimum value of -1.0403E+01, which matches the optimal value for this function. This result indicates the QWGBO algorithm's capability to find the true global minimum, leading to a rank of 1 and demonstrating its outstanding performance.

Table 2 provides the results of the Wilcoxon signed-rank test for each benchmark function. The p-values obtained in the comparisons between QWGBO, and the other algorithms are presented. A significant symbol '+' or ' = ' indicates whether QWGBO statistically outperforms or performs equivalently to the other algorithms. In all comparisons, QWGBO exhibits statistically significant superiority (as indicated by '+') when compared to GBO, GSA, WOA, SMA, and PDO. These results further confirm that the QWGBO algorithm consistently outperforms its counterparts on the benchmark functions, highlighting its robustness and effectiveness in finding optimal solutions. In conclusion, the experimental results clearly demonstrate the superior performance of the QWGBO algorithm in solving the benchmark functions, as it consistently achieves lower minimum values and ranks first in all comparisons. This indicates its potential as an effective optimization algorithm for a wide range of practical applications.

**Table 2. Results of Wilcoxon signed-rank test for benchmark functions.**

| Function | Comparisons | p-value | Significant |
|---|---|---|---|
| $BF_1$ | QWGBO versus GBO | 1.7344E−06 | + |
|  | QWGBO versus GSA | 1.7344E−06 | + |
|  | QWGBO versus WOA | 1.7344E−06 | + |
|  | QWGBO versus SMA | 1.7344E−06 | + |
|  | QWGBO versus PDO | 1.7333E−06 | + |
| $BF_2$ | QWGBO versus GBO | 1.7344E−06 | + |
|  | QWGBO versus GSA | 1.7344E−06 | + |
|  | QWGBO versus WOA | 1.7344E−06 | + |
|  | QWGBO versus SMA | 1.7344E−06 | + |
|  | QWGBO versus PDO | 1.7333E−06 | + |
| $BF_3$ | QWGBO versus GBO | 1.7344E−06 | + |
|  | QWGBO versus GSA | 1.7344E−06 | + |
|  | QWGBO versus WOA | 1.7344E−06 | + |
|  | QWGBO versus SMA | 1.7344E−06 | + |
|  | QWGBO versus PDO | 1.6976E−06 | + |
| $BF_4$ | QWGBO versus GBO | 4.8828E−04 | + |
|  | QWGBO versus GSA | 5.0000E−01 | = |
|  | QWGBO versus WOA | 2.5356E−06 | + |
|  | QWGBO versus SMA | 2.4414E−04 | + |
|  | QWGBO versus PDO | 1.7300E−06 | + |

## AVR system

In this section, we discuss the components and characteristics of the AVR system, which is crucial in maintaining stable voltage levels within a power generation and distribution network. The AVR system plays a pivotal role in regulating and controlling the generator's output voltage. Table 3 provides a comprehensive overview of the critical components of the AVR system, along with their corresponding transfer functions and adopted values [23,25,46–57]. These components include the Amplifier, Exciter, Generator, and Sensor, each contributing to the overall control system.

The amplifier's function is to adjust the input signal to achieve the desired output voltage regulation. The exciter is responsible for generating the necessary field current to control the generator's output voltage. The generator forms the core of the power generation system and produces the electrical energy that requires regulation. The sensor component is responsible for monitoring the system's output and providing feedback for control adjustments. Fig 2 depicts the block diagram of an uncontrolled AVR system, highlighting the key components and their interconnections. In this unregulated state, the system lacks the necessary control mechanisms to maintain voltage stability, leading to potential issues such as voltage fluctuations and instability.

**Table 3. The components of an AVR system along with their transfer functions and adopted values.**

| Component | Transfer function of component | Used values |
|---|---|---|
| Amplifier | $G_a = \frac{K_a}{1+T_a s}$ | $K_a = 10$ and $T_a = 0.1$ s |
| Exciter | $G_e = \frac{K_e}{1+T_e s}$ | $K_e = 1.0$ and $T_e = 0.4$ s |
| Generator | $G_g = \frac{K_g}{1+T_g s}$ | $K_g = 1.0$ and $T_g = 1.0$ s |
| Sensor | $H_s = \frac{K_s}{1+T_s s}$ | $K_s = 1.0$ and $T_s = 0.01$ s |

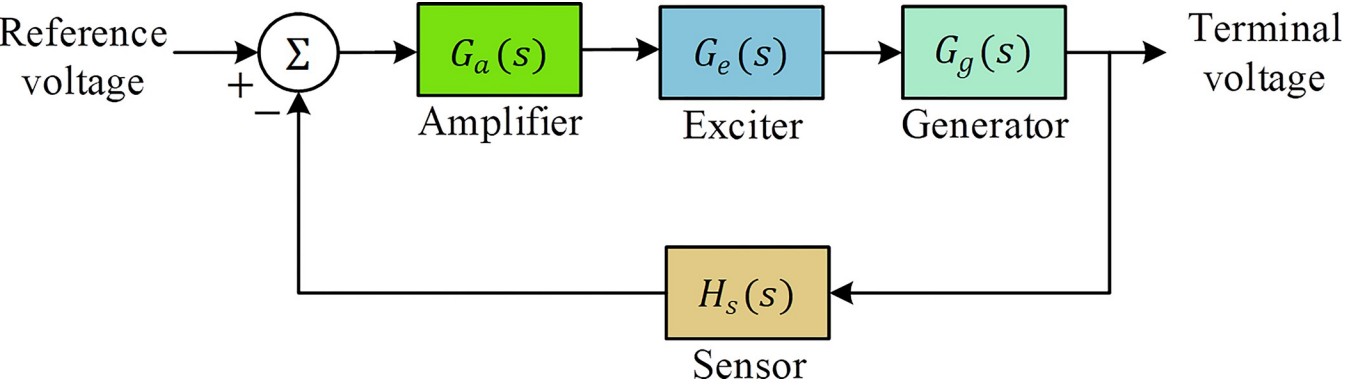

**Fig 2. Block diagram of uncontrolled AVR system.**

Fig 3 illustrates the step response of the uncontrolled AVR system, showing the system's reaction to a sudden change in input conditions. In the absence of voltage regulation, the system's response exhibits undesirable characteristics, underscoring the need for an effective AVR control system. Effective voltage regulation is vital for the reliability and stability of electrical power networks, making the AVR system a critical element in the field of power engineering.

## New methodology for transient stability enhancement

### Proposed Novel Cascaded RPIDD²-FOPI controller

The standard PID controller is widely employed in AVR systems [65]. To enhance its performance an extended version of the PID controller known as PIDD² controller has also been employed in AVR studies [66]. By introducing a second-order derivative term, the PIDD² controller effectively improves the system's phase margin, minimizes steady-state error, and enhances overall stability. However, it's important to note that the derivative term may not be effective in high-frequency domains. This is due to the risk of amplifying control signals with sensor noise, which can negatively impact system performance. To mitigate this issue, a low-pass filter can be added to the derivative term, resulting in the transfer function of the RPIDD² controller as shown in the following equation [67]:

$$C_{RPIDD^2}(s) = k_{p1} + \frac{k_{i1}}{s} + k_{d1}\frac{n_1 s}{s + n_1} + k_{d2}\left(\frac{n_2 s}{s + n_2}\right)^2 \tag{37}$$

where $k_{p1}$, $k_{i1}$, $k_{d1}$, and $k_{d2}$ denote proportional, integral, derivative, and second-order derivative gains, respectively. $n_1$ and $n_2$ represent the filter coefficients. In this work, we have considered the $11^{th}$ order ($N = 5$) Oustaloup's recursive approximation within the frequency range of [0.001, 1000] $rad/s$, which is a commonly used range in fractional-order control applications [68]. In order to increase the performance of the controller, we have also adopted a FOPI controller, as well. The transfer function of the FOPI controller can be presented in the following form [69]:

$$C_{FOPI}(s) = k_{p2} + \frac{k_{i2}}{s^\lambda} \tag{38}$$

where $k_{p2}$, $k_{i2}$ and $\lambda$ are proportional and integral gains and the fractional order of the employed RPIDD² and FOPI

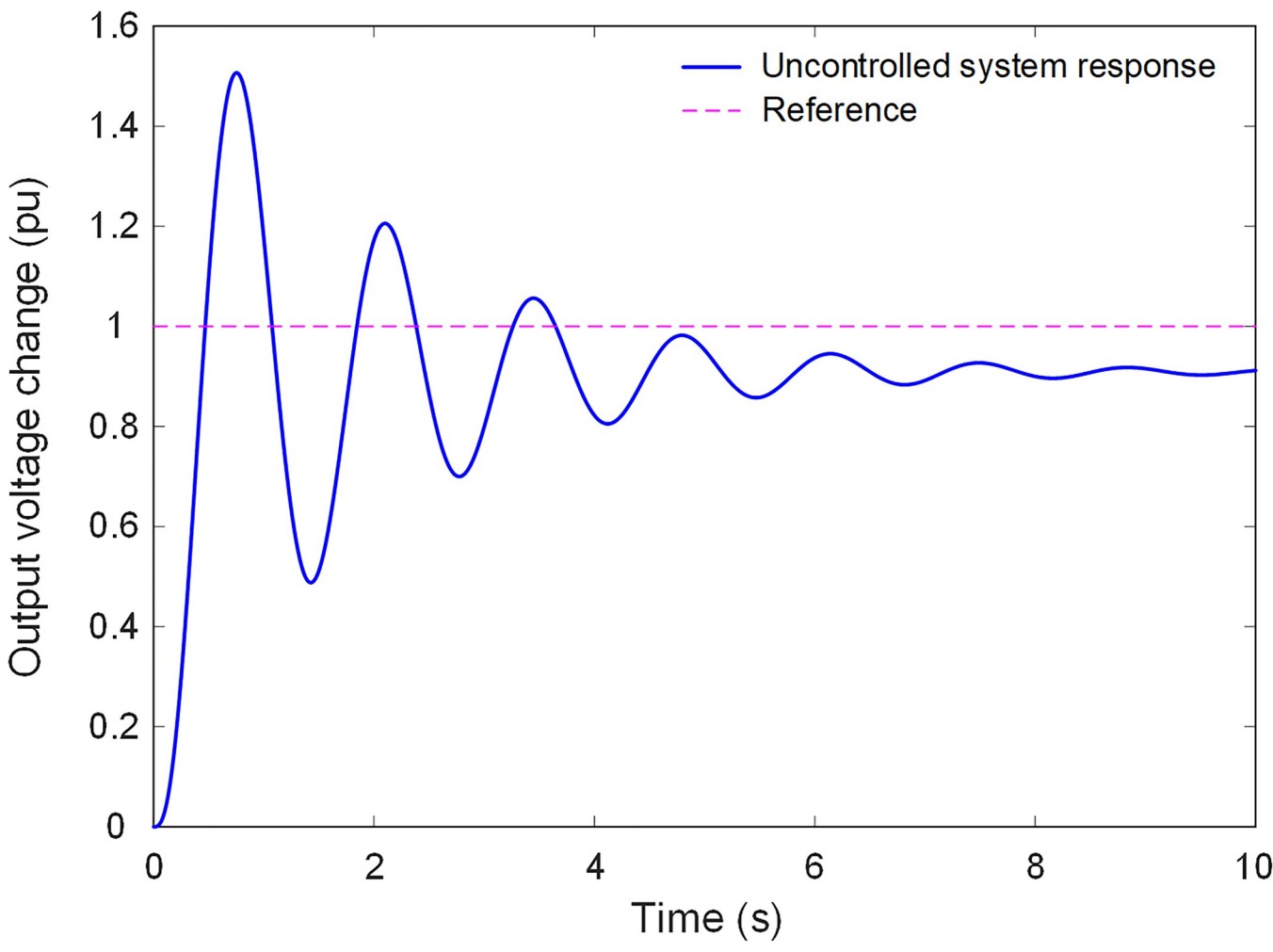

**Fig 3. Step response of uncontrolled system.**

controllers have been interconnected in a cascaded manner which can be defined as follows.

$$C_{RPIDD^2-FOPI}(s) = C_{RPIDD^2}(s) \times C_{FOPI}(s) \qquad (39)$$

The block diagram of the cascaded RPIDD²-FOPI controller is presented in Fig 4 and the block diagram of the AVR system controlled via the proposed cascaded controller is displayed in Fig 5.

### Objective function and constraints of optimization problem

In this study, the following $W$ (Zwe-Lee Gaing's objective function) objective function [65] has been employed for minimization as it can effectively minimize the dynamic response performance criteria (percentage maximum overshoot, steady-state error, settling time and rise time) of the system [70].

$$W = (1 - \psi)(M_{os} + E_{ss}) + \psi(T_{set} - T_{rise}) \qquad (40)$$

In here, $M_{os}$ is the percent overshoot, $T_{set}$ is the settling time, $T_{rise}$ is the rise time, $E_{ss}$ is the steady state error and $\psi$ is a weighting coefficient. The limits for the parameters of the proposed cascaded RPIDD²-FOPI controller are listed in Table 4.

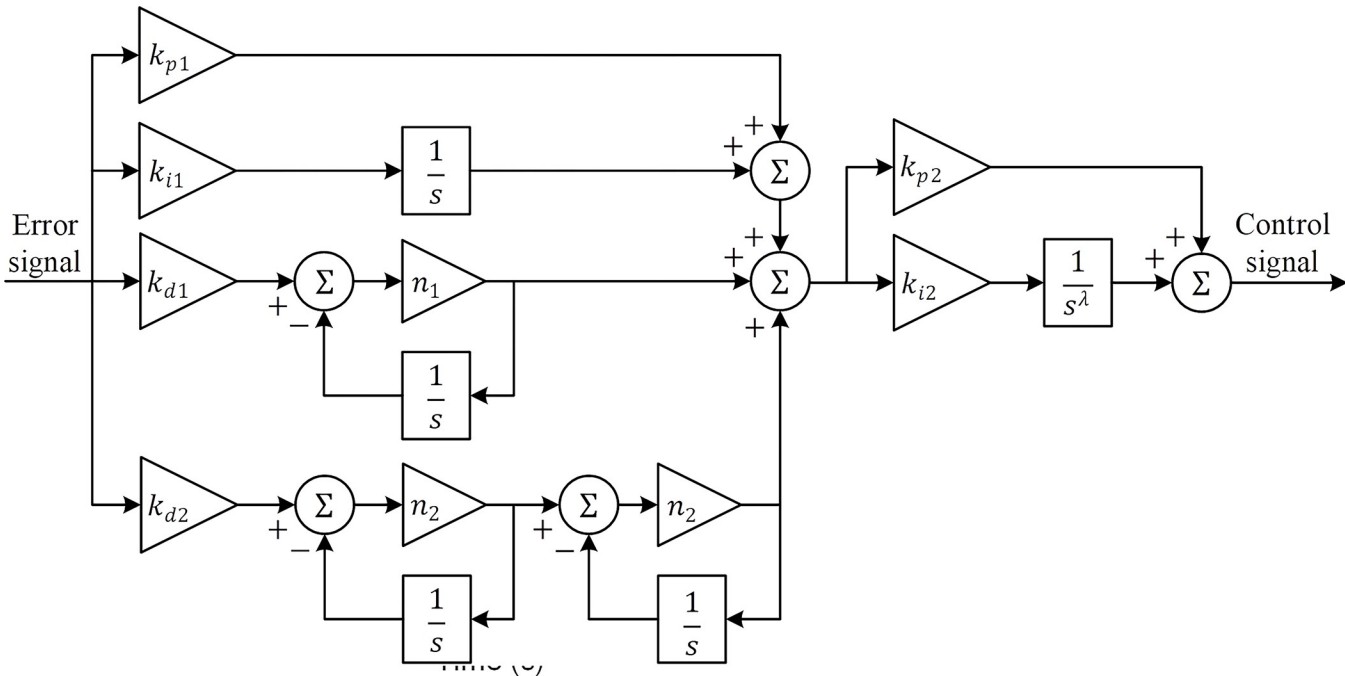

**Fig 4. Block diagram of RPIDD²-FOPI controller.**

## Implementation of QWGBO

The implementation procedure to tune the RPIDD²-FOPI controller using the proposed QWGBO algorithm is illustrated in Fig 6. The related optimization procedure relies on updating the parameters of the system ($k_{p1}$, $k_{i1}$, $k_{d1}$, $k_{p2}$, $k_{i2}$, $k_{d2}$, $n_1$, $n_2$ and $\lambda$) by continuously minimizing the $W$ cost function. For the minimization a total iteration of 50 was chosen with a population size of 30. The algorithm was run independently for 30 times in order to perform the optimization.

## Simulation results and discussion

In this section, we present the comparative performance assessment of the proposed QWGBO algorithm in the context of AVR system control. This comparative assessment involves benchmarking the performance of QWGBO against several competitive optimization algorithms, including original GBO [40], GSA [41], WOA [42], SMA [43] and PDO [44]. These algorithms

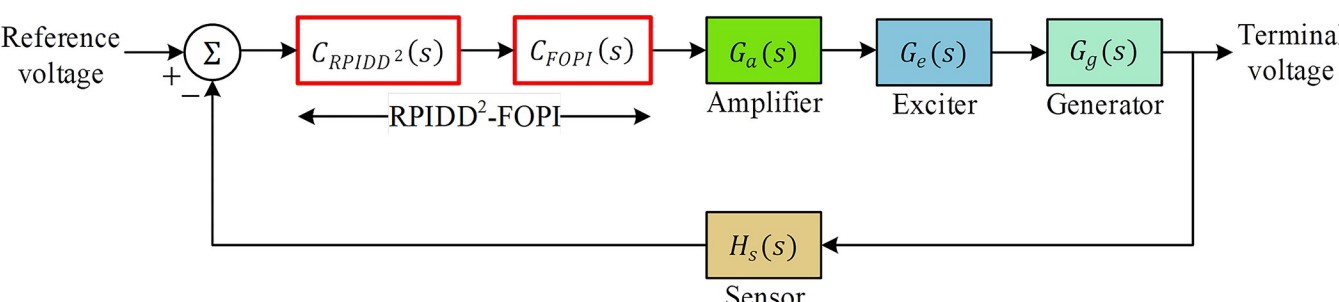

**Fig 5. Block diagram of RPIDD²-FOPI controlled AVR system.**

**Table 4. Employed boundaries for the parameters of cascaded RPIDD²-FOPI controller.**

| Limits | $k_{p1}$ | $k_{i1}$ | $k_{d1}$ | $k_{p2}$ | $k_{i2}$ | $k_{d2}$ | $n_1$ | $n_2$ | $\lambda$ |
|---|---|---|---|---|---|---|---|---|---|
| Lower | 1 | 0.5 | 1 | 0.5 | 0.02 | 0.01 | 10 | 10 | 0.5 |
| Upper | 4 | 4 | 4 | 2 | 1 | 1 | 2000 | 2000 | 1.5 |

have been recognized for their competitive optimization capabilities, making them suitable for performance evaluations. It's important to highlight that the time complexity per run for the utilized algorithms is 42.2621 $s$ for QWGBO, 39.6748 $s$ for GBO, 43.4998 $s$ for GSA, 48.3819 $s$ for WOA, 45.6261 $s$ for SMA and 47.5425 $s$ for PDO. As it is an improved version of GBO with different modifications, it is expected to have higher time complexity, however, this difference is negligible for our proposed QWGBO approach. Compared to other competitors, the proposed approach has lower time complexity, making it superior in this regard.

Table 5 presents the statistical results for the AVR system control, including minimum, maximum, average, standard deviation, median, and rank for each algorithm. Notably, the QWGBO algorithm achieved the lowest minimum value of 5.8743E−03, indicating its superior performance in minimizing the objective function. It ranked first (Rank 1) among all the algorithms, showcasing its effectiveness.

Fig 7 provides a visual representation of the statistical results in Table 5 through boxplots. It is evident that the proposed QWGBO algorithm reaches the lowest objective function value within a lower range compared to the other algorithms, emphasizing its efficiency in AVR system control.

Fig 8 illustrates the evolution of the objective function over iterations. It is clear that the QWGBO algorithm converges to the lowest objective function value as iterations progress, demonstrating its ability to effectively optimize the AVR system. Table 6 presents the results of the Wilcoxon signed-rank test, which assesses the statistical significance of the performance differences between QWGBO and the other algorithms. The 'Significant' column indicates that QWGBO significantly outperforms the other algorithms, as denoted by the '+' symbol.

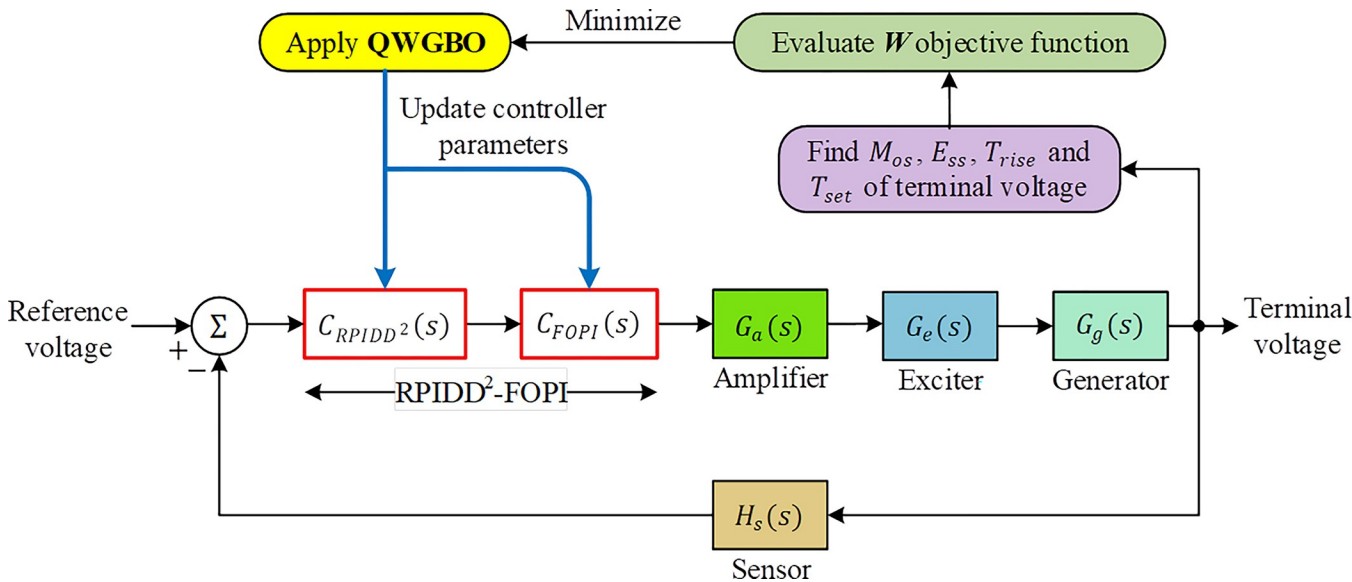

**Fig 6. Recommended novel design method for optimizing AVR system.**

**Table 5. Comparative statistical results on AVR system.**

| Algorithms | Minimum | Maximum | Average | Standard deviation | Median | Rank |
|---|---|---|---|---|---|---|
| QWGBO | **5.8743E−03** | **6.0963E−03** | **5.9811E−03** | **6.9913E−05** | **5.9886E−03** | **1** |
| GBO | 6.3794E−03 | 6.7386E−03 | 6.5311E−03 | 1.0069E−04 | 6.5290E−03 | 2 |
| GSA | 8.3073E−03 | 8.8323E−03 | 8.5008E−03 | 1.2065E−04 | 8.4984E−03 | 6 |
| WOA | 8.2088E−03 | 8.6775E−03 | 8.4020E−03 | 1.1614E−04 | 8.4304E−03 | 5 |
| SMA | 6.6056E−03 | 7.0105E−03 | 6.7341E−03 | 9.3819E−05 | 6.7149E−03 | 3 |
| PDO | 7.3383E−03 | 7.7448E−03 | 7.4974E−03 | 1.1772E−04 | 7.4920E−03 | 4 |

In Table 7, the best controller parameters obtained by each algorithm are listed. These parameters are essential for AVR system control. QWGBO has generated controller parameters that result in a highly competitive performance, further emphasizing its effectiveness.

Fig 9 provides step responses of the AVR system controlled using the parameters obtained by various algorithms and Table 8 lists the comparative transient response performance

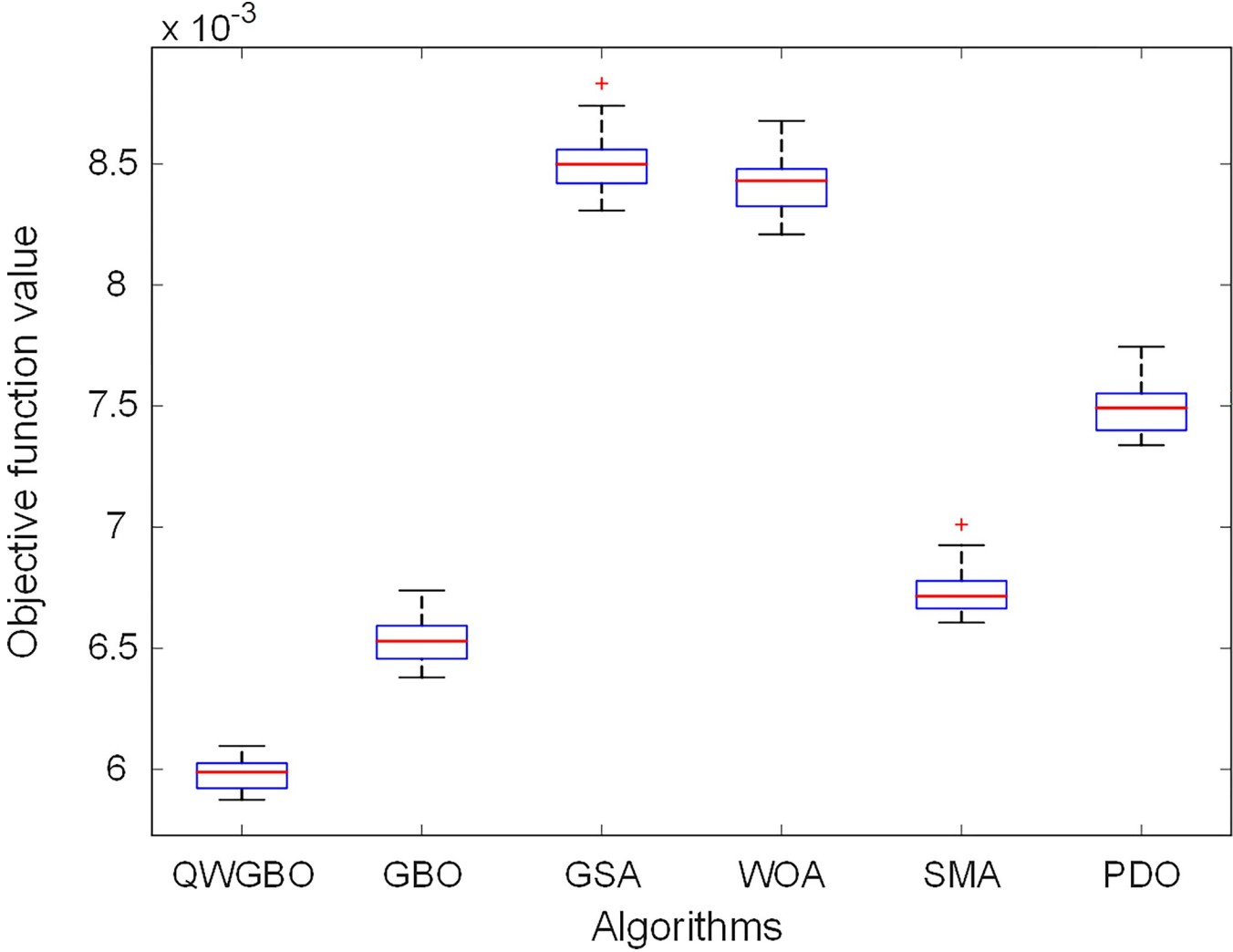

**Fig 7. Boxplots for QWGBO, GBO, GSA, WOA, SMA and PDO algorithms.**

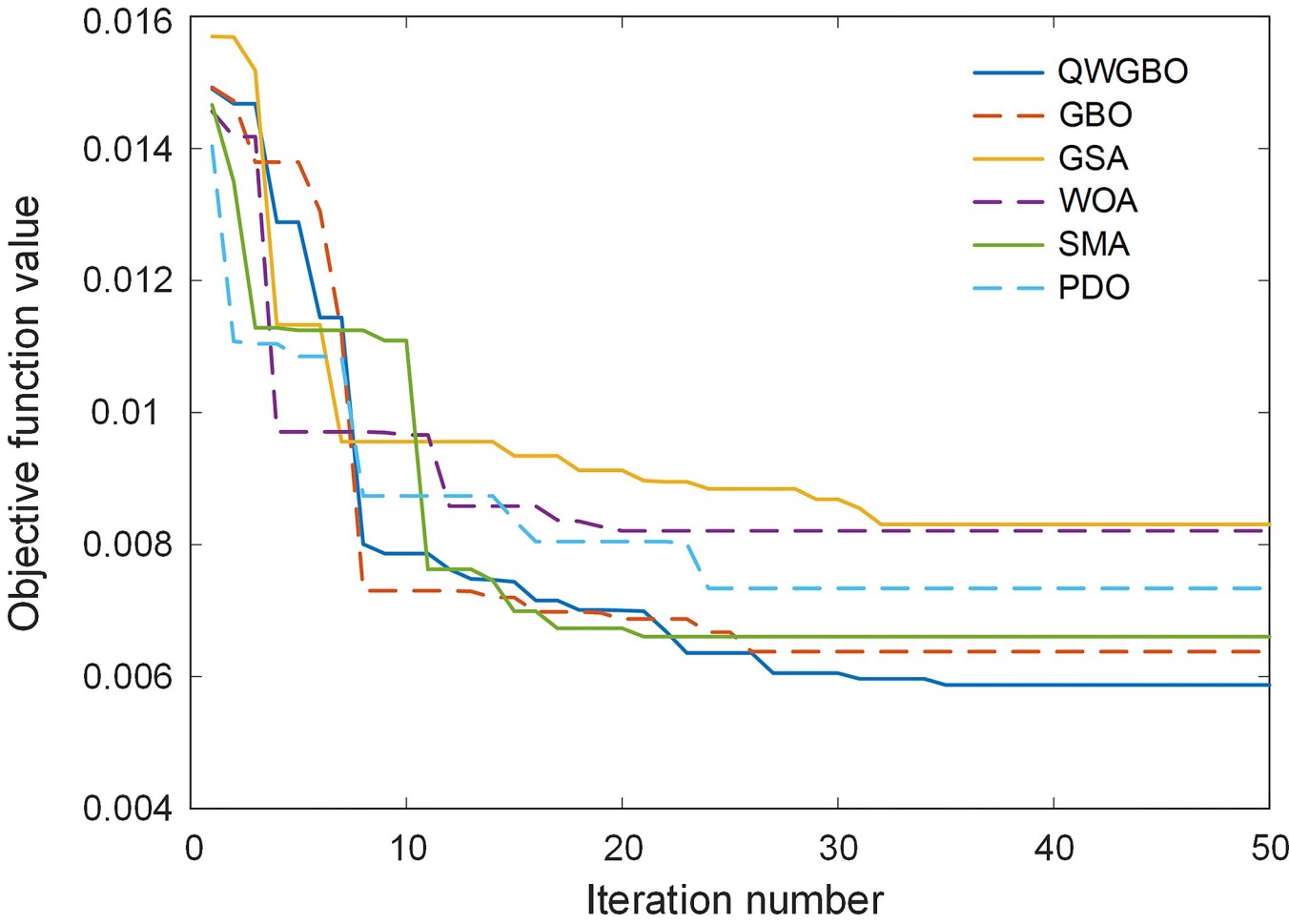

**Fig 8. Evolution of objective function.**

metrics, including rise time, settling time, and overshoot, for each algorithm. QWGBO demonstrates the lowest rise time, settling time, and overshoot, indicating superior transient response characteristics. Fig 10 visually represents the rise time, settling time, and overshoot metrics for different algorithms, providing a clear visualization of the performance comparison. QWGBO's superior performance in these metrics is evident.

To further illustrate the effectiveness of the proposed QWGBO, its performance is examined under varying reference voltages and external disturbances. In this context, Fig 11 presents the step response of the RPIDD²-FOPI controller tuned by QWGBO for the AVR system, considering different reference voltage inputs and external load disturbances. The system's

**Table 6. Results of Wilcoxon signed-rank test for W objective function.**

| Comparisons | p-value | Significant |
|---|---|---|
| QWGBO versus GBO | 1.7344E−06 | + |
| QWGBO versus GSA | 1.7344E−06 | + |
| QWGBO versus WOA | 1.7344E−06 | + |
| QWGBO versus SMA | 1.7344E−06 | + |
| QWGBO versus PDO | 1.7344E−06 | + |

**Table 7. Obtained best controller parameters with different algorithms.**

| Algorithms | $k_{p1}$ | $k_{i1}$ | $k_{d1}$ | $k_{p2}$ | $k_{i2}$ | $k_{d2}$ | $n_1$ | $n_2$ | $\lambda$ |
|---|---|---|---|---|---|---|---|---|---|
| QWGBO | 3.8225 | 1.1128 | 1.2499 | 1.4423 | 0.033033 | 0.10965 | 1570.3 | 1938.3 | 0.86467 |
| GBO | 3.7478 | 1.2208 | 1.3578 | 1.3275 | 0.10060 | 0.11354 | 1057.4 | 1841.7 | 1.3726 |
| GSA | 3.7053 | 0.60785 | 1.3454 | 1.2801 | 0.25807 | 0.11113 | 1194.9 | 1599.3 | 0.86088 |
| WOA | 3.7501 | 0.89175 | 1.4491 | 1.1856 | 0.22197 | 0.11326 | 1565.0 | 1755.8 | 1.1087 |
| SMA | 2.8575 | 1.3745 | 1.3022 | 1.3735 | 0.32483 | 0.10615 | 1480.1 | 1910.5 | 1.0625 |
| PDO | 3.7475 | 1.1525 | 1.4269 | 1.2602 | 0.27124 | 0.11676 | 1251.8 | 1850.6 | 1.3068 |

response demonstrates the capability of the proposed approach to adeptly handle changes in both reference voltage and external load disturbances. Regardless of dynamic variations and load disturbances, the output voltage adeptly tracks the reference inputs, affirming the robustness and adaptability of the proposed method for efficient AVR system operation.

The Bode diagram provides insights into the stability of the open-loop system and the margin of stability. Examining the Bode diagram depicted in Fig 12, numerical results indicate a gain margin of 28.4 *dB* and a phase margin of 70.7 degrees. This analysis is performed to

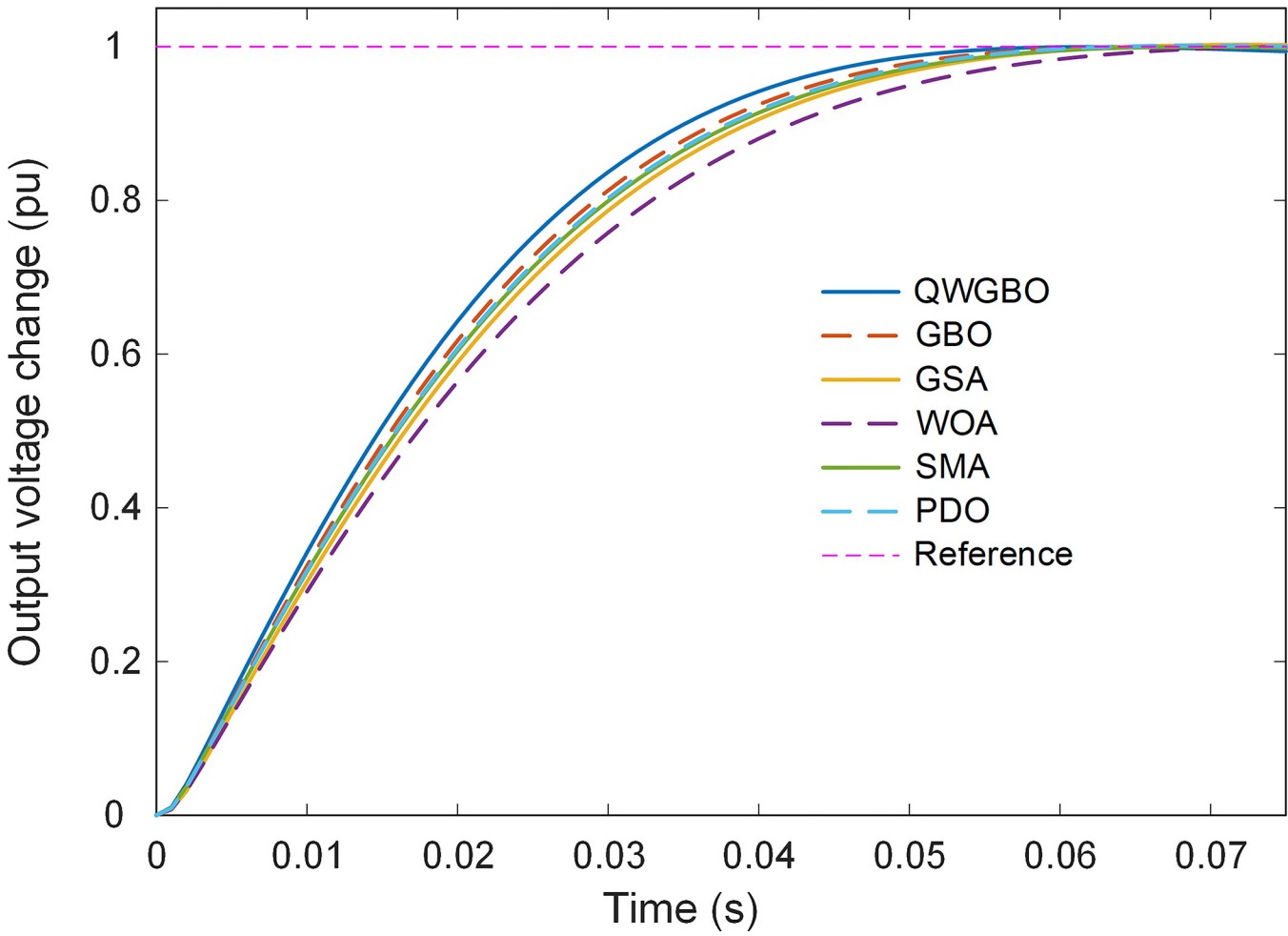

**Fig 9. Step responses of RPIDD²-FOPI controlled AVR system.**

**Table 8. Comparative transient response performance metrics.**

| Algorithms | Rise time (s) | Settling time (s) | Overshoot (%) |
|---|---|---|---|
| QWGBO | 0.0316 | 0.0475 | 0 |
| GBO | 0.0334 | 0.0505 | 0.0139 |
| GSA | 0.0353 | 0.0534 | 0.2623 |
| WOA | 0.0382 | 0.0584 | 0.1227 |
| SMA | 0.0346 | 0.0525 | 0.0067 |
| PDO | 0.0342 | 0.0516 | 0.1478 |

further demonstrate the stability of the proposed approach in terms of the frequency domain. These numerical outcomes affirm that the proposed system design (QWGBO-based RPIDD²-FOPI controlled system) exhibits an excellent frequency response, as well.

Table 9 illustrates the energy and maximum control signal of various controllers, aiming to showcase the comparative effectiveness of controller outputs tuned by diverse algorithms. The table distinctly reveals that the QWGBO-based RPIDD²-FOPI controller proposed in this study exhibits the highest control effort among the controllers. This heightened effort is

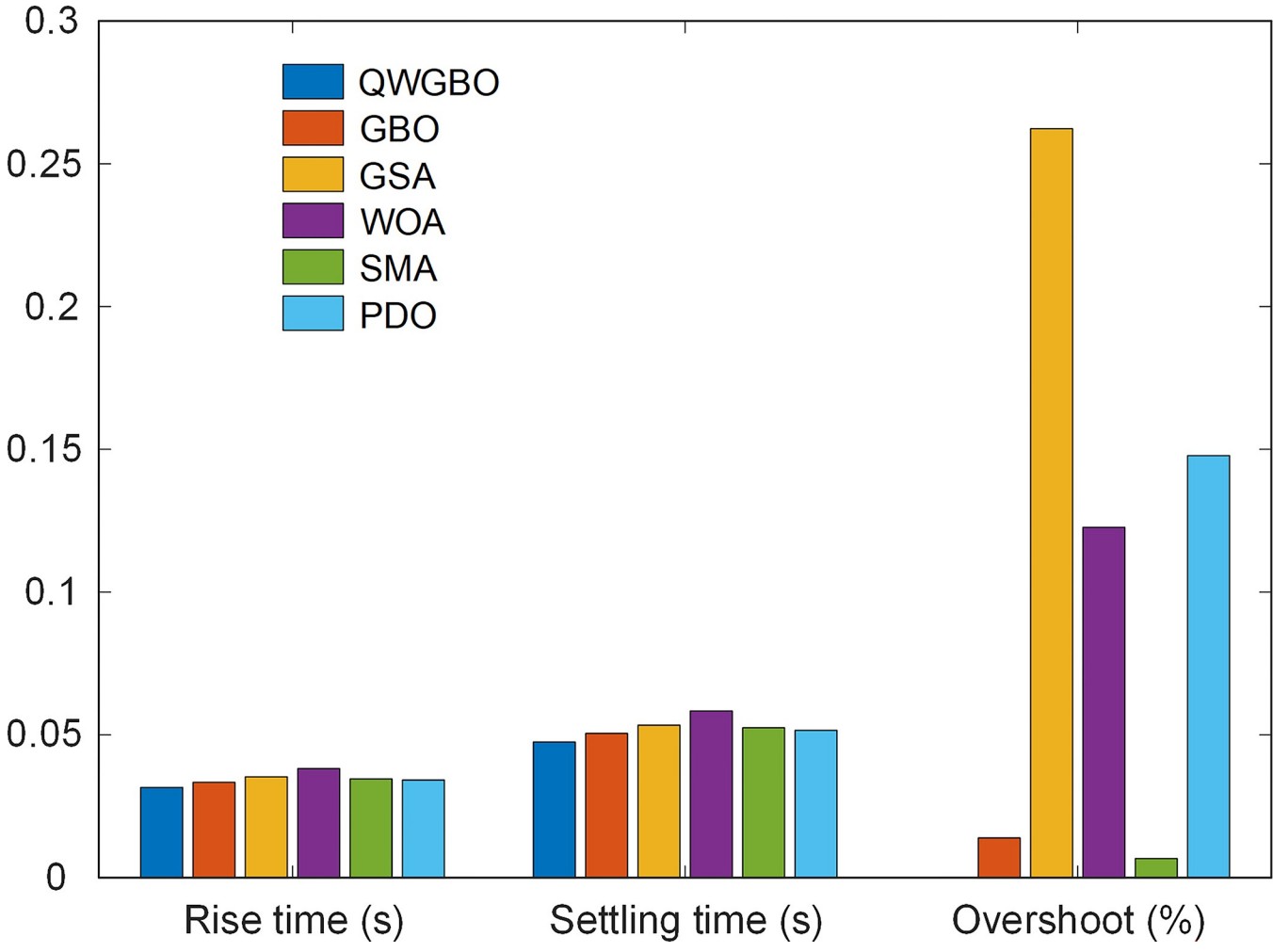

**Fig 10. Visual representation of rise time, settling time and overshoot for different algorithms.**

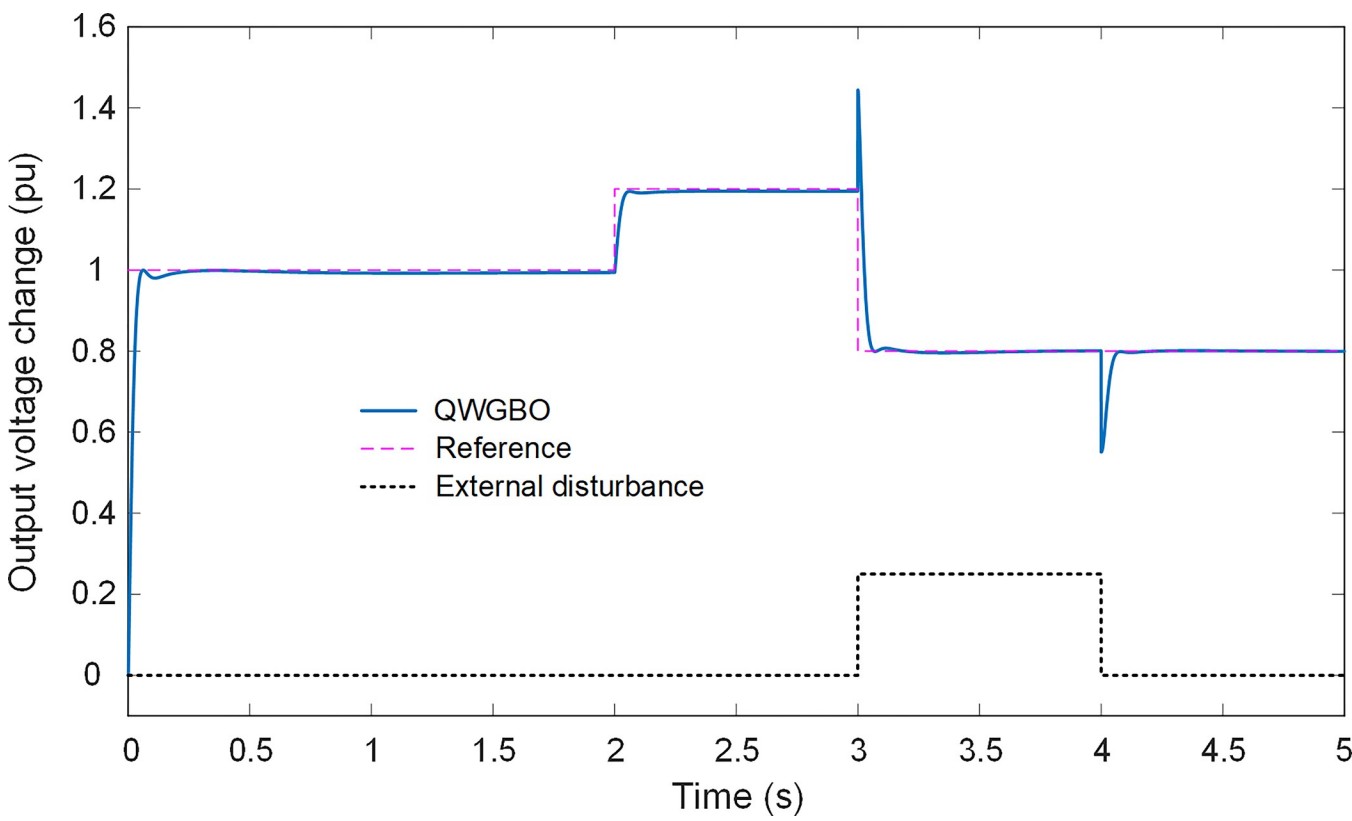

**Fig 11. Step responses of QWGBO based RPIDD²-FOPI controlled AVR system under different reference voltage inputs and external load disturbances.**

attributed to achieving a faster transient response, making the proposed controller the quickest in terms of response time. While the maximum control effort value is slightly elevated, it remains within manageable limits, especially in real-time applications, where it can be constrained using a saturation block. This emphasizes the efficacy of the proposed approach in achieving rapid and dynamic responses.

The robustness of the proposed QWGBO algorithm is evident in Table 10, where comparative analysis under various system uncertainties showcases its superior performance. Considering the uncertainties introduced in the amplifier, exciter, and generator parameters, QWGBO consistently outperforms the other algorithms in terms of critical performance metrics. For amplifier uncertainty, QWGBO demonstrates a shorter rise time (0.0316 s), faster settling time (0.046 s), and lower overshoot (1.2736%) compared to its counterparts, such as GBO, GSA, WOA, SMA, and PDO. This trend is observed across different uncertainties, emphasizing QWGBO's resilience in handling parameter variations. In the case of exciter uncertainty, QWGBO continues to exhibit superior performance with a shorter rise time (0.037 s), faster settling time (0.0616 s), and minimal overshoot (0.0571%) compared to other algorithms. Notably, QWGBO's efficiency in managing uncertainties is evident in its ability to maintain stability while minimizing overshoot. Under generator uncertainty, QWGBO again outshines other algorithms, achieving a shorter rise time (0.0315 s), faster settling time (0.0473 s), and lower overshoot (0.0908%). This result underscores QWGBO's adaptability and effectiveness in mitigating the impact of parameter uncertainties, ensuring a stable and well-controlled response.

In addition to the comparative assessment against the algorithms used in the above analyses, further comparisons were made with recently reported optimization techniques in the

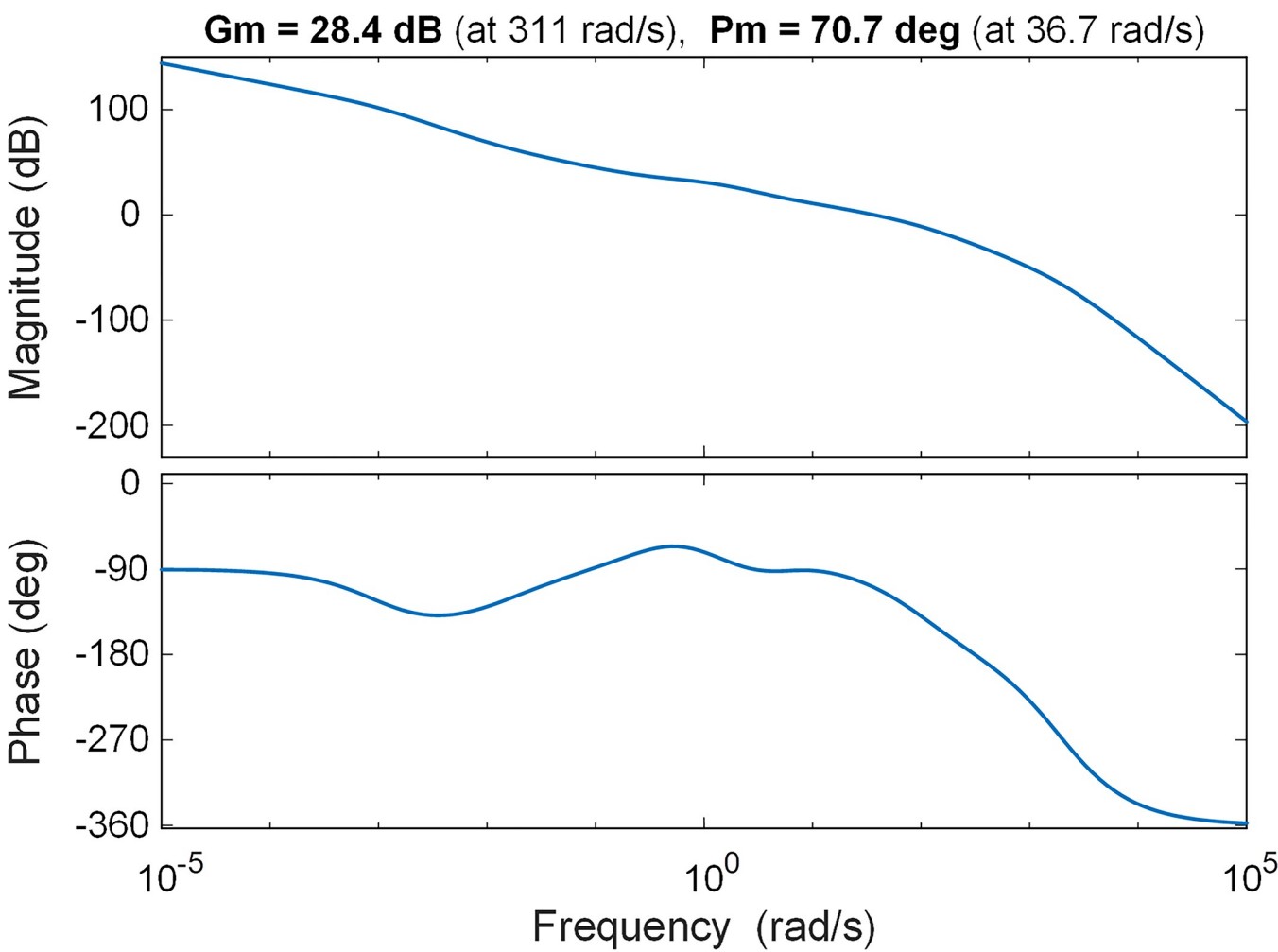

**Fig 12. Bode diagram of QWGBO based RPIDD²-FOPI controlled open-loop AVR system.**

literature (See Table 11). This techniques include hybrid atom search particle swarm optimization (h-ASPSO) based PID controller [46], improved marine predators algorithm (MP-SEDA)-tuned FOPID controller [47], modified artificial bee colony (IABC) based LOA-FOPID [48], equilibrium optimizer (EO) based TI$^\lambda$DND$^2$N$^2$ [23], whale optimization algorithm (WOA) based PIDA [49], symbiotic organism search (SOS) algorithm-based PID-F controller [50], mayfly optimization algorithm based PI$^{\lambda 1}$I$^{\lambda 2}$D$^{\mu 1}$D$^{\mu 2}$ controller [25], Levy flight improved Runge-Kutta optimizer (L-RUN) based PIDD$^2$ controller with master/slave

**Table 9. Energy and maximum control signal of controllers.**

| Algorithms | Energy | $U_{max}$ |
|---|---|---|
| QWGBO | 1.0939E+09 | 4.1392E+05 |
| GBO | 6.7437E+08 | 3.8655E+05 |
| GSA | 1.6334E+08 | 2.8586E+05 |
| WOA | 4.1342E+08 | 3.5143E+05 |
| SMA | 8.7359E+08 | 3.8938E+05 |
| PDO | 7.5048E+08 | 4.0166E+05 |

**Table 10. Comparative robustness analysis for parameter uncertainties.**

| System type | Algorithms | Rise time (s) | Settling time (s) | Overshoot (%) |
|---|---|---|---|---|
| Amplifier uncertainty ($K_a$ = 10.75 and $T_a$ = 0.11 s) | QWGBO | 0.0316 | 0.0460 | 1.2736 |
| | GBO | 0.0334 | 0.0486 | 1.3929 |
| | GSA | 0.0352 | 0.0513 | 1.7307 |
| | WOA | 0.0379 | 0.0554 | 1.6612 |
| | SMA | 0.0345 | 0.0503 | 1.3770 |
| | PDO | 0.0341 | 0.0497 | 1.5640 |
| Exciter uncertainty ($K_e$ = 0.95 and $T_e$ = 0.42 s) | QWGBO | 0.0370 | 0.0616 | 0.0571 |
| | GBO | 0.0390 | 0.0642 | 0 |
| | GSA | 0.0411 | 0.0662 | 0 |
| | WOA | 0.0443 | 0.0715 | 0 |
| | SMA | 0.0404 | 0.0662 | 0.0566 |
| | PDO | 0.0399 | 0.0647 | 0 |
| Generator uncertainty ($K_g$ = 1.05 and $T_g$ = 1.05 s) | QWGBO | 0.0315 | 0.0473 | 0.0908 |
| | GBO | 0.0333 | 0.0502 | 0.1337 |
| | GSA | 0.0352 | 0.0530 | 0.3885 |
| | WOA | 0.0381 | 0.0579 | 0.2553 |
| | SMA | 0.0345 | 0.0521 | 0.0864 |
| | PDO | 0.0342 | 0.0513 | 0.2698 |

approach [51], particle swarm optimization based 2DOF-PI controller with amplifier feedback [52], modified artificial rabbits optimizer (m-ARO) based FOPIDD² controller [53], genetic algorithm (GA) based fuzzy PID controller [54], sine-cosine algorithm (SCA) based FOPID controller with fractional filter [55], imperialist competitive algorithm (ICA) based gray PID controller [56], Rao algorithm based multi-term FOPID controller [57], whale optimization algorithm (WOA) based 2DOF-FOPI [58], chaotic yellow saddle goatfish algorithm (C-YSGA)

**Table 11. Comparisons with the recently reported good works in the literature.**

| Reference | Algorithm | Controller type | Rise time (s) | Settling time (s) | Overshoot (%) |
|---|---|---|---|---|---|
| Proposed | QWGBO | Cascaded RPIDD²-FOPI | **0.0316** | **0.0475** | **0** |
| [46] | h-ASPSO | PID | 0.3097 | 0.4679 | 1.2476 |
| [47] | MP-SEDA | FOPID | 0.083 | 0.1103 | 0.56 |
| [48] | IABC | LOA-FOPID | 0.1373 | 0.3129 | 2.3323 |
| [23] | EO | $TI^\lambda DND^2N^2$ | 0.03752 | 0.0596 | 0.4128 |
| [49] | WOA | PIDA | 0.328 | 0.453 | 2 |
| [50] | SOS | PID-F | 0.267 | 0.371 | 0.007 |
| [25] | MA | $PI^{\lambda 1}I^{\lambda 2}D^{\mu 1}D^{\mu 2}$ | 0.0323 | 0.0500 | **0** |
| [51] | L-RUN | PIDD² with master/slave approach | 0.0357 | 0.0537 | **0** |
| [52] | PSO | 2DOF-PI with amplifier feedback | 0.690 | 3.442 | 2.224 |
| [53] | m-ARO | FOPIDD² | 0.0330 | 0.0493 | **0** |
| [54] | GA | Fuzzy PID | 0.1857 | 0.2963 | 1.0407 |
| [55] | SCA | FOPID with fractional filter | 0.1230 | 0.1670 | 0.1262 |
| [56] | ICA | Gray PID | 0.2305 | 0.3193 | 1.23 |
| [57] | Rao | Multi-term FOPID | 0.0965 | 0.170 | 0.01 |
| [58] | WOA | 2DOF-FOPI | 1.12 | 1.74 | 1.17 |
| [59] | C-YSGA | FOPID | 0.1347 | 0.2 | 1.89 |
| [31] | CSA | FOPI | 2.8829 | 7.347 | 3.6782 |

based FOPID controller [59] and crow search algorithm (CSA) based FOPI controller [31]. The results indicate that the QWGBO algorithm outperforms several state-of-the-art optimization methods, demonstrating its effectiveness in AVR system control. In conclusion, the simulation results and discussions highlight the superior performance of the proposed QWGBO algorithm in optimizing the AVR system. Its effectiveness is demonstrated through lower objective function values, excellent convergence, and competitive controller parameters. Furthermore, QWGBO outperforms other recent optimization techniques, emphasizing its potential in real-world control systems applications. The visual representations in figures and the data in tables collectively provide comprehensive insights into the algorithm's performance and its advantages over alternative optimization methods.

## Conclusion and potential future works

This study introduced an innovative approach to AVR control, focused on enhancing robustness and efficiency. In this regard, the QWGBO is developed as a novel optimizer which improves upon the GBO by introducing exploration and exploitation enhancements, making it particularly effective in addressing complex and high-dimensional optimization problems. The algorithm combines the QIM and the WMS, striking a balance between exploration and exploitation. The effectiveness of the QWGBO algorithm was thoroughly demonstrated through extensive tests using benchmark functions, indicating its superior optimization capabilities. Comparative assessments against various optimization algorithms, including recent techniques, underlined the algorithm's performance, positioning it as a promising solution for power system control and engineering optimization. In the context of AVR control, QWGBO was coupled with a newly proposed cascaded RPIDD²-FOPI controller, promising precision, stability, and rapid response. This novel approach was validated through several assessments (statistical, boxplot, convergence profile, Wilcoxon signed-rank test, transient and frequency responses, performance against varying input reference and external load disturbance, controller effort and robustness). The results affirmed the effectiveness of the QWGBO-tuned cascaded RPIDD²-FOPI controller, demonstrating its superior performance compared to existing control and optimization techniques. In terms of future research directions, several promising avenues emerge. Future investigations may explore QWGBO's potential in optimizing other power infrastructure components, thereby enhancing overall system efficiency and reliability. Additionally, future studies could delve into the customization of RPIDD²-FOPI controller and QWGBO approach for specific power system requirements, potentially leading to tailored solutions that optimize performance in diverse contexts. Furthermore, the integration of advanced cost functions, alternative optimization techniques, and innovative controller designs may provide avenues for further enhancing AVR systems.

## Author Contributions

**Conceptualization:** Mohammad Salman.

**Data curation:** Mohammad Salman.

**Formal analysis:** Serdar Ekinci, Václav Snášel, Davut Izci, Mohammad Salman.

**Funding acquisition:** Mohammad Salman, Ahmed A. F. Youssef.

**Investigation:** Ahmed A. F. Youssef.

**Methodology:** Václav Snášel, Davut Izci.

**Software:** Serdar Ekinci, Rizk M. Rizk-Allah.

**Supervision:** Davut Izci.

**Validation:** Rizk M. Rizk-Allah, Ahmed A. F. Youssef.

**Writing – original draft:** Serdar Ekinci, Václav Snášel, Mohammad Salman, Ahmed A. F. Youssef.

**Writing – review & editing:** Serdar Ekinci, Václav Snášel, Rizk M. Rizk-Allah, Davut Izci.

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
