## [Decision Letter · Decision Letter 0]

3 Jan 2024

PONE-D-23-39168Optimizing AVR System Performance via a Novel Cascaded RPIDD2-FOPI Controller and QWGBO ApproachPLOS ONE

Dear Dr. Shukri Salman,

Thank you for submitting your manuscript to PLOS ONE. After careful consideration, we feel that it has merit but does not fully meet PLOS ONE’s publication criteria as it currently stands. Therefore, we invite you to submit a revised version of the manuscript that addresses the points raised during the review process.

We look forward to receiving your revised manuscript.

Kind regards,

Yogendra Arya

Academic Editor

PLOS ONE

[This work has been supported by an internal grant project of VSB-Technical University of Ostrava (SGS project, grant number SP 2023/076).]

 [The author(s) received no specific funding for this work.]

5. Please include the reference section of your manuscript.

Reviewers' comments:

Reviewer's Responses to Questions

**Comments to the Author**

1. Is the manuscript technically sound, and do the data support the conclusions?

Reviewer #1: Yes

Reviewer #2: Yes

Reviewer #3: Partly

Reviewer #4: Yes

2. Has the statistical analysis been performed appropriately and rigorously? 

Reviewer #1: Yes

Reviewer #2: Yes

Reviewer #3: No

Reviewer #4: Yes

3. Have the authors made all data underlying the findings in their manuscript fully available?

Reviewer #1: Yes

Reviewer #2: Yes

Reviewer #3: Yes

Reviewer #4: Yes

4. Is the manuscript presented in an intelligible fashion and written in standard English?

Reviewer #1: Yes

Reviewer #2: Yes

Reviewer #3: Yes

Reviewer #4: Yes

5. Review Comments to the Author

Reviewer #1: Appreciating and publishable work but requires few improvements. The paper should be further revised by incorporating the following comments:

1. The main contributions of this paper should be further summarized and clearly demonstrated. This reviewer suggests the authors exactly mention what is new compared with existing approaches and why the proposed approach is needed to be used instead of the existing methods.

2. Introduction should be divided the into 6 subsections: 1) Background, 2) Literature review, 3) Research gap and motivation, 4) Challenges, 5) Contribution (give in points) and 6) paper organization.

3. Robustness of the method should also be validated in frequency/time domain.

4. Compare computation burdens of different methods.

5. There are few typo and grammatical errors in the paper.

6. Add few more results in different operating conditions.

7. Figure 1, should not be in image format.

8. In RPIDD2-FOPI controller, what is the filter order and lower/up frequency range regarding FOC.

9. Literature review should be more strengthen by adding few more papers like https://doi.org/10.1080/15567036.2022.2156637;
https://doi.org/10.1016/j.jestch.2020.12.023;
https://doi.org/10.1016/j.seta.2022.102671;

https://doi.org/10.1016/j.ref.2022.09.006.

10. Conclusion should be focused. It is too lengthy.

Reviewer #2: The submitted manuscript presents a meta-heuristic approach based RPIDD2-FOPI control method for terminal voltage regulation. The submitted work needs to be revised as per the following comments:

1. Significant amount of literature is present in the area of Automatic Voltage Regulators; hence authors are advised to include a critical literature review on AVR systems. And based on the research gaps, justify the novelty and contributions of the presented work.

2. Authors have used QWGB optimization approach for the optimal tuning of RPIDD2-FOPI parameters in the AVR system, which may be treated as a significant contribution of this work. In this regard, I suggest the authors to highlight its advantage over recently developed meta heuristic approaches.

3. Add more analysis for AVR systems, such as parameter sensitivity, and external disturbance.

4. Prove the stability of the proposed controller using linear stability analysis methods.

Reviewer #3: Though paper claims better results, there are some concerns as below.

1. QWGBO-tuned cascaded RPIDD2-FOPI controller - has nine tuning parameters. In comparison with other methods, they have less tuning parameters and simple structure like FOPI and PID. What is possible that due to more degree of freedom, slight improvement in results indicated. Thus, it is not fully accepted.

2. Can this method compare with other FOPI type or FOPID or 2DOF type in AVR system? with same initial condition and objective function?

3. It is required to check the controller output signal and measures of efforts with others. It should be less.

4. Include some other authors' papers on related FOPI(d) and AVR.

5. Stability analysis should be included with proposed control structure.

Reviewer #4: 1. Well defined objective and the structure of your manuscript is good.

2. Using QWGBO on AVR system is innovative and the Figure:9 clearly depicts QWGBO giving lowest objective function value compared to other search algorithms.

3. Figure:10 and Table 8 clearly shows that your designed controller with QWGBO algorithm is performing well compared to other control algorithms in terms of rise time, overshoot and settling time.

4. Robustness of your proposed algorithm is not presented in the paper , as a controller it has to show robustness for disturbances and even for system parameters change. So, better include the controller performance with QWGBO when the system parameters are changed between 5-10%, also compare the results with other algorithms.

5. Also , better check the proposed controller performance with different reference inputs.

6. PLOS authors have the option to publish the peer review history of their article (what does this mean?). If published, this will include your full peer review and any attached files.

Reviewer #1: No

Reviewer #2: **Yes: **Vineet kumar

Reviewer #3: No

Reviewer #4: **Yes: **G.Suri babu

---

## [Author Response · Author response to Decision Letter 0]

25 Jan 2024

Response to Reviewers

We would like to extend our heartfelt appreciation to the reviewers and the editorial team for their positive and constructive comments on our manuscript. Their valuable feedback and dedicated efforts have played a crucial role in enhancing the quality of our work. The constructive nature of their feedback has helped us refine our ideas and strengthen the clarity of our presentation. We would like to express our gratitude to all individuals involved in the review process for their valuable and constructive comments. In response to the reviewers' comments, we have carefully considered each suggestion and incorporated appropriate revisions into the manuscript. Thank you all for your time, effort, and valuable input throughout the review process.

Reviewer #1:

Overall Comment: Appreciating and publishable work but requires few improvements. The paper should be further revised by incorporating the following comments:

Response: We sincerely appreciate your positive evaluation of our work and your recognition of its originality and suitability for publication in the journal. We have carefully considered your comments and have performed a thorough revision to address the areas that require further improvement.

Comment 1: The main contributions of this paper should be further summarized and clearly demonstrated. This reviewer suggests the authors exactly mention what is new compared with existing approaches and why the proposed approach is needed to be used instead of the existing methods.

Response: We have updated the entire manuscript in order to better demonstrate the contribution of this work. We have especially re-designed the structure of the introduction section in order to clearly demonstrate the contributions of this work. 

Comment 2: Introduction should be divided the into 6 subsections: 1) Background, 2) Literature review, Comment 3) Research gap and motivation, 4) Challenges, 5) Contribution (give in points) and 6) paper organization.

Response: We have updated the introduction section accordingly.

Comment 3: Robustness of the method should also be validated in frequency/time domain.

Response: We have included robustness analysis in terms of parameter sensitivity accordingly.

Comment 4: Compare computation burdens of different methods.

Response: We have now included a relevant statement within the simulation section regarding the computational burdens of the compared methods. 

Comment 5: There are few typo and grammatical errors in the paper.

Response: We have thoroughly checked the manuscript and corrected the existing typos and grammatical errors accordingly.

Comment 6: Add few more results in different operating conditions.

Response: We have increased the number of analyses accordingly. We have included time complexity results, robustness analysis, step responses for different reference inputs, additional stability analysis, and controller effort analysis accordingly. Please see details in the simulation section.

Comment 7: Figure 1, should not be in image format.

Response: We have included the content of the figure as a table and also referred it as Algorithm 1 instead of Figure 1.

Comment 8: In RPIDD2-FOPI controller, what is the filter order and lower/up frequency range regarding FOC.

Response: In this work, we have considered the 11th order (N=5) Oustaloup’s recursive approximation within the frequency range of [0.001, 1000] rad/s, which is a commonly used range in fractional-order control applications. We have now included a relevant statement within the “New methodology” section and provided the relevant information accordingly. The filter coefficients have been optimized in this study, and the best results obtained for each algorithm are listed in Table 7.

Comment 9: Literature review should be more strengthen by adding few more papers like https://doi.org/10.1080/15567036.2022.2156637;
https://doi.org/10.1016/j.jestch.2020.12.023;
https://doi.org/10.1016/j.seta.2022.102671;
https://doi.org/10.1016/j.ref.2022.09.006.

Response: We have now included the suggested references into the manuscript accordingly in order to strengthen the literature review.

Comment 10: Conclusion should be focused. It is too lengthy.

Response: We have updated the conclusion section accordingly.

Reviewer #2:

Comment 1: Significant amount of literature is present in the area of Automatic Voltage Regulators; hence authors are advised to include a critical literature review on AVR systems. And based on the research gaps, justify the novelty and contributions of the presented work.

Response: We have now updated the manuscript by including a dedicated literature review section. We have also included research gap, challenges and contribution sections in order to address the comment of the esteemed reviewer.

Comment 2: Authors have used QWGB optimization approach for the optimal tuning of RPIDD2-FOPI parameters in the AVR system, which may be treated as a significant contribution of this work. In this regard, I suggest the authors to highlight its advantage over recently developed meta heuristic approaches.

Response: We have updated the manuscript in order to highlight the advantage of our proposed method over other metaheuristics.

Comment 3: Add more analysis for AVR systems, such as parameter sensitivity, and external disturbance.

Response: We have included the relevant analyses accordingly.

Comment 4: Prove the stability of the proposed controller using linear stability analysis methods.

Response: We have shown the transient response stability previously. After considering the comment of the esteemed reviewer, we have included the stability for the frequency response, as well.

Reviewer #3:

Comment 1: QWGBO-tuned cascaded RPIDD2-FOPI controller - has nine tuning parameters. In comparison with other methods, they have less tuning parameters and simple structure like FOPI and PID. What is possible that due to more degree of freedom, slight improvement in results indicated. Thus, it is not fully accepted.

Response: The RPIDD2-FOPI controller proposed in our study is a pioneering contribution in the literature for AVR system. Despite its inclusion of nine parameters, the proposed controller structure does not impose a significant computational burden on the system, while elevating its stability to the highest levels. For instance, PID controllers, with only three parameters, have shown limitations in enhancing stability performance as desired. This led researchers to propose controllers with more parameters, such as FOPID [46] (with 5 parameters), PIDA [48] (with 6 parameters), TIλDND2N2 [23] (with 8 parameters), and PIλ1Iλ2Dµ1Dµ2 [25] (with 9 parameters) for AVR systems. Controllers with fewer parameters, like PID, have been outperformed by these alternatives in terms of effectiveness. Furthermore, when comparing the stability performance of the AVR system, the proposed QWGBO-based controller structure consistently yields superior results compared to all approaches in the literature. This is clearly evident in the comparative tables and figures presented in the article.

Comment 2: Can this method compare with other FOPI type or FOPID or 2DOF type in AVR system? with same initial condition and objective function?

Response: We have included the following references and compared the performance accordingly.

- Padiachy, V., Mehta, U., Azid, S., Prasad, S., & Kumar, R. (2022). Two degree of freedom fractional PI scheme for automatic voltage regulation. Engineering Science and Technology, an International Journal, 30, 101046.

- Micev, M., Ćalasan, M., & Oliva, D. (2020). Fractional order PID controller design for an AVR system using Chaotic Yellow Saddle Goatfish Algorithm. Mathematics, 8(7), 1182.

- Bhullar, A. K., Kaur, R., & Sondhi, S. (2022). Optimization of fractional order controllers for AVR system using distance and levy-flight based crow search algorithm. IETE Journal of Research, 68(5), 3900-3917

Comment 3: It is required to check the controller output signal and measures of efforts with others.

Response: We have included the controller efforts for each approach accordingly.

Comment 4: Include some other authors' papers on related FOPI(d) and AVR.

Response: We have included the following related references accordingly.

- Padiachy, V., Mehta, U., Azid, S., Prasad, S., & Kumar, R. (2022). Two degree of freedom fractional PI scheme for automatic voltage regulation. Engineering Science and Technology, an International Journal, 30, 101046.

- Micev, M., Ćalasan, M., & Oliva, D. (2020). Fractional order PID controller design for an AVR system using Chaotic Yellow Saddle Goatfish Algorithm. Mathematics, 8(7), 1182.

- Bhullar, A. K., Kaur, R., & Sondhi, S. (2022). Optimization of fractional order controllers for AVR system using distance and levy-flight based crow search algorithm. IETE Journal of Research, 68(5), 3900-3917

- Khan, I. A., Alghamdi, A. S., Jumani, T. A., Alamgir, A., Awan, A. B., & Khidrani, A. (2019). Salp swarm optimization algorithm-based fractional order PID controller for dynamic response and stability enhancement of an automatic voltage regulator system. Electronics, 8(12), 1472.

- Silas, M., & Bhusnur, S. (2023). Optimal Robust Controller Design for a Reduced Model AVR System Using CDM and FOPIλDμ. In Robotics, Control and Computer Vision: Select Proceedings of ICRCCV 2022 (pp. 297-311). Singapore: Springer Nature Singapore.

- Sivanandhan, A., & Thriveni, G. (2024). Optimal design of controller for automatic voltage regulator performance enhancement: a survey. Electrical Engineering, 1-16.

Comment 5: Stability analysis should be included with proposed control structure.

Response: We have both shown the transient response and frequency response stability analyses accordingly.

Reviewer #4:

Comment 1: Well defined objective and the structure of your manuscript is good.

Response: Thank you for acknowledging the well-defined objective and the positive feedback on the manuscript's structure. Your encouragement is greatly appreciated.

Comment 2: Using QWGBO on AVR system is innovative and the Figure 9 clearly depicts QWGBO giving lowest objective function value compared to other search algorithms.

Response: We appreciate your recognition of the innovation in using QWGBO on the AVR system.

Comment 3: Figure 10 and Table 8 clearly shows that your designed controller with QWGBO algorithm is performing well compared to other control algorithms in terms of rise time, overshoot and settling time.

Response: Many thanks for highlighting the positive performance of the designed controller with the QWGBO algorithm.

Comment 4: Robustness of your proposed algorithm is not presented in the paper, a controller it has to show robustness for disturbances and even for system parameters change. So, better include the controller performance with QWGBO when the system parameters are changed between 5-10%, also compare the results with other algorithms.

Response: We have included robustness analysis accordingly.

Comment 5: Also, better check the proposed controller performance with different reference inputs.

Response: We have included the relevant analyses accordingly.

---

## [Decision Letter · Decision Letter 1]

5 Feb 2024

Optimizing AVR system performance via a novel cascaded RPIDD2-FOPI controller and QWGBO approach

PONE-D-23-39168R1

Dear Dr. Shukri Salman,

We’re pleased to inform you that your manuscript has been judged scientifically suitable for publication and will be formally accepted for publication once it meets all outstanding technical requirements.

Kind regards,

Yogendra Arya

Academic Editor

PLOS ONE

Additional Editor Comments (optional):

Reviewers' comments:

Reviewer's Responses to Questions

**Comments to the Author**

1. If the authors have adequately addressed your comments raised in a previous round of review and you feel that this manuscript is now acceptable for publication, you may indicate that here to bypass the “Comments to the Author” section, enter your conflict of interest statement in the “Confidential to Editor” section, and submit your "Accept" recommendation.

Reviewer #1: All comments have been addressed

Reviewer #2: All comments have been addressed

Reviewer #3: All comments have been addressed

Reviewer #4: All comments have been addressed

2. Is the manuscript technically sound, and do the data support the conclusions?

Reviewer #1: Yes

Reviewer #2: Yes

Reviewer #3: Yes

Reviewer #4: Yes

3. Has the statistical analysis been performed appropriately and rigorously? 

Reviewer #1: Yes

Reviewer #2: Yes

Reviewer #3: Yes

Reviewer #4: Yes

4. Have the authors made all data underlying the findings in their manuscript fully available?

Reviewer #1: Yes

Reviewer #2: Yes

Reviewer #3: Yes

Reviewer #4: Yes

5. Is the manuscript presented in an intelligible fashion and written in standard English?

Reviewer #1: Yes

Reviewer #2: Yes

Reviewer #3: Yes

Reviewer #4: Yes

6. Review Comments to the Author

Reviewer #1: No more comments.

Authors have incorporated all the suggestions. All comments have been addressed.

The paper may be accepted

Reviewer #2: Authors have successfully addressed my comments and concerns. I am happy to recommend the manuscript for publication in its current form.

Reviewer #3: (No Response)

Reviewer #4: 1.As per the recommendations, robust analysis is incorporated by considering uncertainties in the amplifier, exciter, and generator. Results from Table 10 depict that the controller based on the QWGBO algorithm is performing well.

2.Figure 11 shows the AVR output tracking the reference input with QWGBO-based controller .

7. PLOS authors have the option to publish the peer review history of their article (what does this mean?). If published, this will include your full peer review and any attached files.

Reviewer #1: **Yes: **Pankaj Dahiya

Reviewer #2: **Yes: **Dr Vineet Kumar

Reviewer #3: **Yes: **Utkal Mehta

Reviewer #4: **Yes: **G Suri babu
